# **Assessing Inter-Model Agreement in Convection-Permitting Simulations of Extreme Winds for Wind Energy Applications**

Nathalia Correa-Sánchez<sup>1</sup>, Xiaoli Guo Larsén<sup>2</sup>, Eleonora Dallan<sup>1</sup>, Marco Borga<sup>1</sup>, and Francesco Marra<sup>3</sup>

Correspondence: Nathalia Correa-Sánchez (nathalia.correasanchez@phd.unipd.it)

## Abstract.

Convection-permitting models (CPMs) have great potential for wind energy applications such as wind energy planning, turbine design loads, and operational safety in a climate change context. In fact, compared to coarse-resolution models, they have an improved representation of atmospheric processes and surface characteristics that directly influence winds at heights relevant for turbines. Evaluating how well CPMs perform at reproducing extreme wind events is crucial for wind energy applications. Inter-model comparisons provide insights into uncertainties and enhance the credibility of CPM-based applications. Here, we use a new framework to examine the agreement among three CPMs from the CORDEX Flagship Pilot Study in simulating extreme wind speeds in central Europe. This framework combines surface-based spatial categorisation with Principal Component Analysis and with the non-asymptotic Simplified Metastatistical Extreme Value (SMEV) method, able to estimate rare return levels, such as the 50-year wind speeds  $U_{50}$  required for wind turbine design, from the short CPM simulations. Our results show large agreement between the models, with the first Principal Component explaining 74.2% of the total variance and indicating a strong consensus in extreme wind patterns, despite systematic differences in magnitudes. Stronger agreement emerges during the winter, when extreme winds are driven by synoptic conditions, and less concordance during summer, when localised convective events cause most extremes. Our research emphasises the value of using CPM ensembles over single-model assessments of extreme winds, and provides the wind energy community with baseline information on CPM capabilities and limitations in estimating wind speed extremes.

#### 1 Introduction

Wind energy is a key part of Europe's shift to renewable energy, providing nearly 19% of the continent's electricity (European Commission, 2025). A successful wind energy development requires an accurate assessment of extreme wind speeds at turbine hub heights, typically around 100 m, as they are essential for project planning, turbine design loads, and operational safety considerations (Ma et al., 2022). Climate models have served as an effective tool for these applications by enabling wind

<sup>&</sup>lt;sup>1</sup>Department of Land Environment Agriculture and Forestry, University of Padova, Padova, Italy.

<sup>&</sup>lt;sup>2</sup>Department of Wind Energy, Technical University of Denmark, Roskilde, Denmark.

<sup>&</sup>lt;sup>3</sup>Department of Geosciences, University of Padova, Padova, Italy.

30

55

resource assessments across diverse geographical regions, providing spatially consistent datasets for extreme value analysis, and supporting long-term planning under changing climate scenarios (Prein et al., 2015; Archer and Jacobson, 2005). In climate studies, multi-model ensembles are the primary means of obtaining accurate estimates of model uncertainties (Prein et al., 2015; Fosser et al., 2024). The regional variability in extreme wind changes under future scenarios exhibits intensification of extreme winds in some regions and decreases in others, underscoring the importance of using high-resolution climate modelling approaches for accurate wind energy impact assessments under changing climate conditions (Pryor et al., 2020; Outten and Sobolowski, 2021; Ma et al., 2025).

In recent years, global and regional climate models (GCMs and RCMs) have served as essential tools for understanding large-scale atmospheric flows. Although these models reproduce general wind patterns, their coarse resolution (~10 km) cannot capture surface heterogeneity and key processes like convection that affect local wind behaviour and extremes (Soares et al., 2012; Prein et al., 2015; Prein, 2023). Convection-Permitting Models (CPMs) provide high-resolution simulations with a grid spacing of less than 4 km where deep convection is explicitly resolved instead of being parameterised (Prein et al., 2015; Schär et al., 2020), thereby offering an alternative to the limitations of GCMs and RCMs. Originally developed to improve the representation of future extreme precipitation, CPMs also bring added potential benefits for other atmospheric variables, such as winds at 100 m. In fact, they enable a more accurate representation of surface characteristics and their interactions with surface winds, and they explicitly resolve processes of gust front propagation and downdrafts associated with convective storms (Manning et al., 2022; Prein, 2023), and also preserve the natural spectral characteristics of the wind in the high mesoscale frequency range (Correa-Sánchez et al., 2025b).

The CORDEX Flagship Pilot Study (CORDEX-FPS) framework on convective phenomena (Coppola et al., 2020) developed the first multi-model ensemble of convection-permitting simulations in the Euro-Mediterranean region. Its simulations are generated for historical conditions as well as near- and far-future projections, eventually offering opportunities for medium- and long-term planning. Previous research that relied on the wind speed field from CPMs has focused mainly on average conditions (Ruiz et al., 2022; Ma et al., 2022), rather than on extreme wind speed. While Chen et al. (2024) showed significant improvements using CPMs compared to coarser resolution simulations for renewable energy assessment, methodological challenges emerge from other studies. For example, Frisius et al. (2024) recognises the biases that may be introduced by vertical extrapolation when working only with simulations of winds at 10 m, and focuses on the changes in the average wind energy metrics instead of extreme events. Ma et al. (2025), instead, used return level estimations based on threshold-exceedances to study destructive winds in Canada. However, this work is based on relatively few exceedance samples, which may introduce important uncertainties in the subsequent extrapolations. In part, the extreme value extrapolations with CPMs are limited by the short simulations available, typically on the order of 10 years. This occurs because CPMs are computationally expensive simulations (Prein et al., 2015), and conducting the multi-decadal simulations needed for extreme value analysis is highly costly.

Traditionally, the estimation of extreme wind return levels has been based on well-established statistical frameworks rooted in asymptotic assumptions of Extreme Value Theory (EVT), mainly using the Generalised Extreme Value (GEV) distribution for annual maxima approaches and the Generalised Pareto Distribution (GPD) for peaks-over-threshold methodologies (such as

90

the one by Ma et al., 2025). These approaches are extensively documented in European wind climatology (Donat et al., 2010), analysis of synoptic-scale circulation patterns affecting wind extremes (Pfahl, 2014), and analyses of mixed wind climates (Harris and Cook, 2014). However, as demonstrated by Larsén et al. (2015) in their analysis of uncertainties in 50-year wind extreme estimates ( $U_{50}$ ), both GEV and GPD methods exhibit substantial sensitivity to record length. This arises because parameter estimation becomes increasingly unreliable when based on short time series, reflecting the influence of sampling uncertainties (Coles et al., 2001). These considerations become even more critical when considering future climate scenarios and diverse terrain conditions, underscoring the need for tools that provide detailed information essential for wind energy development and methods to adequately estimate extreme wind speeds from short simulation records.

An effective use of CPMs for wind energy involves key methodological considerations: evaluating model performance and uncertainties, employing suitable statistical methods for extreme value estimation from model time series, and accounting for spatial variability in surface characteristics. Here, we address the limitations of observational data through an inter-model comparison, which we achieve using Principal Component Analysis (PCA) to decompose the inter-model variance structure into consensus signals and divergences. To address the short time series issue, the Simplified Metastatistical Extreme Value (SMEV) approach (Marra et al., 2019) is used because it considers the full distribution of the independent events instead of only focusing on annual maxima or threshold exceedances, making it more reliable for estimating return levels from limited CPM time series (Dallan et al., 2024). SMEV has been successfully used for the assessment of extreme precipitation probabilities from in-situ observations as well as remotely sensed data and CPM simulations (Miniussi and Marra, 2021; Dallan et al., 2023; Correa-Sánchez et al., 2025a; Marra et al., 2022). Here, we implement SMEV for the first time in wind speed analysis, extending this non-asymptotic extreme value framework to wind energy applications. Our approach is then refined by recognising surface heterogeneity as a factor that modulates surface winds. Since surface characteristics change boundary layer dynamics through differential heating, roughness effects, and turbulence generation (Courault et al., 2007; Avissar and Schmidt, 1998) affecting wind patterns at 100 m height, a spatial categorisation method allows the systematic assessment of CPMs across different surface-atmosphere interaction regimes that are relevant to wind energy applications.

Inter-model agreement assessment is especially important when examining CPMs, because there are several ways to implement the model schemes and resolve convection (Coppola et al., 2020; Abramowitz et al., 2019), which can result in differences in how they represent wind fields, providing an indicator of prediction uncertainty, and recognising that the consensus among different models reflects consistency rather than absolute accuracy (Bellucci et al., 2015). Notably, these schemes may lead to diverse representations of wind fields over surfaces with different characteristics. Considering these factors, ensemble approaches have proven effective in assessing model uncertainty, as they use a set of individual models to detect model-specific biases and reduce overall prediction errors (Yan and Tang, 2013). Assessing the agreement of multi-model ensembles provides information about the ability to distinguish between robust signals (where models agree) and uncertain predictions (where models diverge), thereby offering insight into the reliability of the simulated patterns (Yan and Tang, 2013).

This study aims to estimate extreme wind events using SMEV through an assessment of the agreement among three CPMs, establishing a framework for wind energy applications. To do so, we structured an approach with four key objectives. First, we examine the inter-model differences in wind extremes within the domain, exploring the variation in extreme winds represent

tation among the three CPMs. Second, we establish spatial categories based on surface roughness, climate, and topographic features, providing a basis for evaluation and comparison of analyses based on spatial categories, accounting for the heterogeneous surface-atmosphere interactions that modulate wind patterns. Third, we assess the spatio-temporal agreement among three CPMs of the CORDEX-FPS project in simulating wind speeds at turbine height across central Europe. Fourth, we apply the SMEV approach to wind speed data for obtaining estimates of extreme wind speeds from short model simulations. We then discuss how CPM ensemble approaches provide added value for wind extremes and may contribute to long-term wind energy planning.

## 100 2 Study area and data

105

110

115

120

The study domain extends from  $0.5^{\circ}$ E to  $16.3^{\circ}$ E in longitude and from  $40.2^{\circ}$ N to  $49.7^{\circ}$ N in latitude, covering approximately 250,000 km² in central Europe (Figure 1). The region encompasses considerable topographical diversity, ranging from the Mediterranean sea, the flat plains of the Po Valley and northern France, to elevations exceeding 4,000 m a.s.l. in the Alpine region. It spans a range of climatic conditions, from Mediterranean influences in the south to continental regimes in the north. There are also complex land-sea interactions along the Mediterranean and Adriatic coastlines within this domain, providing varied surface characteristics that modulate wind patterns at turbine-relevant heights, creating a suitable scenario for testing CPM performance across different wind generation mechanisms.

We focus on wind speed at 100 m height, as it represents the international standard for wind turbines (Davis et al., 2023; National Renewable Energy Laboratory, 2023; Petersen, 1997). We analyse data from three CPM members of the CORDEX-FPS (Coppola et al., 2020) for which such a variable was made available at the time of the study (Table 1). In particular, wind speeds were derived directly from the zonal (u 100 m) and meridional (v 100 m) wind components, available at hourly frequency, for the reanalysis period of 2000-2009. In all models, ERA-Interim (Dee et al., 2011) provides consistent initial and lateral boundary conditions, ensuring that inter-model differences arise from model physics and stochastic variability only, with no contribution from the boundary forcing. Intermediate-resolution regional climate models (RCMs, with a 12 km horizontal resolution) are nested within ERA-Interim, with the CPMs subsequently nested within these RCMs. The three models employed are based on fundamentally different atmospheric cores and convection schemes, providing diverse representations of sub-grid scale processes critical for wind simulation at hub heights. This diversity in model structure and parameterisations makes them particularly suitable for assessing inter-model agreement and characterising model uncertainty. To enable direct comparison, all CPM outputs were remapped onto a common 3 km regular grid covering the study domain using a bilinear interpolation. This method preserves the spatial continuity of the wind field while keeping the values within a plausible physical range.

**Table 1.** CPM members used for inter-model agreement assessment, showing reference names, original resolutions, and coupled RCM configurations. All models were remapped to a common 3 km grid for comparison.

| Institute                             | CPM                                    | RCM                                   |  |
|---------------------------------------|----------------------------------------|---------------------------------------|--|
| CMCC                                  | CCLM                                   | CCLM                                  |  |
| Euro-Mediterranean Center on Climate  | 3 km (Adinolfi et al., 2020; Rockel    | 12 km (Adinolfi et al., 2020; Rockel  |  |
| Change                                | et al., 2008)                          | et al., 2008)                         |  |
| CNRM                                  | CNRM-AROME41t1                         | CNRM-ALADIN63                         |  |
| Centre National de Recherches         | 2.5 km (Caillaud et al., 2021)         | 12 km (Nabat et al., 2020)            |  |
| Météorologiques                       |                                        |                                       |  |
| ЕТН                                   | COSMO-crCLIM                           | COSMO-crCLIM                          |  |
| Institute for Atmospheric and Climate | 2.2 km (Leutwyler et al., 2016; Rockel | 12 km (Leutwyler et al., 2017; Rockel |  |
| Science                               | et al., 2008)                          | et al., 2008)                         |  |

#### 3 Methods

We describe here our analysis of the CPM ensemble, which combines a spatial stratification into climate-roughness-topography categories (Section 3.1), quantification of inter-model agreement and divergences using Principal Component Analysis (PCA) and seasonal correlations (Section 3.2), and estimation of extreme wind speeds using SMEV (Section 3.3).

#### 125 3.1 Spatial categorisation

In order to understand how inter-model differences in wind extremes at 100 m relate to surface characteristics and provide a spatial comparison context for subsequent analyses, we develop a spatial categorisation based on climate patterns, roughness, and topography.

On the one hand, climate classification systems capture this heterogeneity, as different Köppen-Geiger climate zones exhibit different surface wind characteristics due to contrasting thermal regimes, land-ocean distributions, and seasonal circulation patterns that modify boundary layer processes across spatial scales (Beck et al., 2023; Ulbrich et al., 2009). On the other hand, Petersen (1997) showed that wind power meteorology needs to consider other surface-atmosphere interactions that influence superficial wind flows, such as surface roughness effects and orographic acceleration. Orographic lifting and thermal circulation modify local wind patterns (Smith, 1979), while surface roughness controls boundary layer dynamics and turbulent mixing (Kaimal and Finnigan, 1994), including the modulation of convective processes (Avissar and Schmidt, 1998; Kang and Bryan, 2011). An analogous categorisation was implemented in the European Wind Atlas methodology (Troen and Lundtang Petersen, 1989) and further developed with modern multivariate categorisation schemes in the New European Wind Atlas (Dörenkämper et al., 2020).

#### 3.1.1 Climate

Climate is the primary classification category, because extreme winds strongly depend on large-scale atmospheric flows, circulation weather types, North Atlantic Oscillation (NAO) phases, cyclone tracks, mean sea level pressure patterns, and pressure gradients (Donat et al., 2010; Pfahl, 2014). We use the Köppen-Geiger climate classification by Beck et al. (2023), which represents the 1991-2020 period at high resolution (1 km). Köppen-Geiger climate zones reflect large-scale atmospheric circulation patterns that also influence regional wind characteristics. After remapping to a 3 km resolution to match the CPM common grid, we obtained 13 distinct climate types in our study domain. To ensure statistical robustness and a more straightforward interpretation of the results, we grouped these 13 climate types into four levels: Arid (Ar), Temperate (Tm), Cold (Co), and Tundra (Td), as shown in Fig. 1a. This aggregation maintains the primary atmospheric differences that drive extreme wind variability across the domain.

#### 3.1.2 Roughness

Surface roughness at a particular location is modulated by local terrain characteristics, with surface roughness influencing how wind speed varies with height, as described by models such as the logarithmic wind profile (Stull, 1988; Petersen, 1997); greater roughness causes larger wind retardation near the ground. Surface roughness categorisation was based on the roughness length  $(z_0)$  data that has been used as input to the COSMO model, and was derived from the methodology of Demuzere et al. (2008), which associates roughness length values with land cover types from the CORINE Land Cover Map (European Environment Agency, 2020). This approach aligns with the roughness classification framework established in the European Wind Atlas (Troen and Lundtang Petersen, 1989), ensuring consistency with established wind energy assessment practices. We defined five roughness levels ranging from  $R_1$  (very smooth) to  $R_5$  (very rough), representing the spectrum from water surfaces and open terrain to dense urban areas and forests (Fig. 1b). Detailed information of the roughness levels is in Table 2.

## 3.1.3 Topography

Topography has a significant influence on wind flow near the surface and up to heights relevant for wind turbines. Topographic patterns cause wind acceleration near ridge crests and deceleration in the valleys (Jackson and Hunt, 1975; Troen and Lundtang Petersen, 1989). The slope variance can be used as a metric for terrain complexity, able to capture the presence of significant elevation changes (Riley et al., 1999), indicating areas where these acceleration/deceleration effects are most likely and pronounced.

We use the ETOPO 2022 global relief dataset at 15 arc-second resolution (MacFerrin et al., 2025), which was remapped to 3 km resolution to match the CPM common grid. Using terrestrial elevation values (> 0 m), we calculated slope for each grid cell and subsequently computed slope variance by comparing each cell's slope against its eight neighbouring cells within a 3×3 moving window. To objectively define topographic complexity levels, we applied the Jenks natural breaks optimisation algorithm (Chen et al., 2013), which iteratively partitions the slope variance data into classes that minimise within-class variance while maximising between-class variance, thus identifying natural clustering boundaries in the data distribution. This

**Table 2.** Roughness classes based on European Wind Atlas with logarithmic scale of aerodynamic roughness length  $(z_0)$ . Categories range from very smooth surfaces  $(R_1)$  to very rough terrain  $(R_5)$ , following standard wind energy assessment protocols.

| Roughness Level              | Start $z_0$ [m] | End $z_0$ [m] | Centroid  | <b>Description (Examples)</b>           |
|------------------------------|-----------------|---------------|-----------|-----------------------------------------|
|                              |                 |               | (approx.) |                                         |
| R <sub>1</sub> : Very smooth | 0               | 0.0003        | 0.0001    | Very smooth surfaces (water, ice, fresh |
|                              |                 |               |           | snow)                                   |
| R <sub>2</sub> : Smooth      | 0.0003          | 0.03          | 0.003     | Smooth surfaces (short grass,           |
|                              |                 |               |           | compacted snow, ploughed fields)        |
| R <sub>3</sub> : Moderate    | 0.03            | 0.3           | 0.1       | Moderately rough surfaces (grasslands,  |
|                              |                 |               |           | crops, vineyards)                       |
| R <sub>4</sub> : Rough       | 0.3             | 0.7           | 0.5       | Rough surfaces (low forests, urban      |
|                              |                 |               |           | areas with low scattered buildings)     |
| R <sub>5</sub> : Very rough  | 0.7             | $\infty$      | > 1       | Very rough surfaces (dense forests,     |
|                              |                 |               |           | urban centres with tall buildings)      |

classification yielded four topographic levels:  $T_1$  (Flat),  $T_2$  (Gentle),  $T_3$  (Moderate), and  $T_4$  (Complex), representing increasing degrees of terrain complexity, as shown in Fig. 1c.

## 3.1.4 Combined Classification

The three categorical layers defined above (climate, roughness, topography) were combined pixel-by-pixel to obtain 52 possi-52 possi-53 ble total category combinations (Figure 1d). Each grid point receives an identifier composed of the level of each of the three classification layers, representing a unique environmental signature. This layer combination generates composite codes following the format [Climate level][Roughness level][Topography level], where each component retains its original categorical designation. For example,  $ArR_3T_4$  represents a location characterised by arid climate (Ar), moderate surface roughness  $(R_3)$ , and complex topographic conditions  $(T_4)$ .

The spatial distribution of the combined categories reveals distinct clustering patterns that reflect the underlying surface heterogeneity in the domain. Inner-continental areas exhibit the greatest diversity of spatial categories, ranging from moderate-roughness and flat terrain in the northern plains to high-roughness mountainous regions in the Alps and other elevated areas. Coastal and marine environments are represented by specific categories that capture land-sea transitions and varying degrees of surface roughness associated with different coastal morphologies.

## 185 3.1.5 Stratified random sampling

The stratified random sampling is a sampling technique that incorporates two key principles: firstly, stratified sampling ensures the representation of all spatial categories, with random selection that mitigates spatial autocorrelation and clustering effects;




**Figure 1.** Spatial categories and their respective levels a) Climate, b) Roughness c) Topography. A combination of the 3 layers in panel d) with 52 total resulting categories within the domain.

secondly, equal number selection was applied to ensure that each spatial category contributed an identical number of time series, thereby providing equal analytical weight regardless of the category's spatial extent within the domain.

Given the need to focus on predominant representative conditions, a frequency analysis was conducted to identify the most prevalent spatial combinations. The categories were ranked by their abundance within the domain, and the 17 characteristic spatial categories that collectively represent more than 97.7% of the domain were identified. These combinations maintain adequate representation of the diverse geographic conditions of the region (Figure 2) while ensuring that rare or transitional combinations with limited spatial extent do not bias the extremes CPMs assessment.

Spatial randomisation was implemented within each of the dominant categories to mitigate potential spatial autocorrelation effects and sampling bias that could arise from clustered or systematic point selection. Here, we selected 100 points per category to provide a statistically representative sample size while maintaining computational feasibility for subsequent analyses. This approach was employed to extract representative wind speed time series from each spatial category, ensuring statistical comparability across diverse geographical conditions. This allows the subsequent statistical analyses and performance metrics to be directly comparable across all spatial categories, enabling a robust assessment of CPM behaviour under varying surface-atmosphere interaction regimes.

The random selection yields a total of 5,100 complete time series covering the entire 10-year period (17 categories  $\times$  3 models  $\times$  100 points) and representing the various surface characteristics within the study domain (Figures 1 and 2). The

selected locations are shown in Fig. 1d, while the statistical distribution of all spatial categories, and the most frequent ones, can be found in Fig. 2.

**Figure 2.** Spatial category distribution and filtering strategy. a) Distribution of pixel relative frequencies across all 52 spatial categories ranked by abundance. Blue bars indicate categories above the 25th percentile threshold (n=17), red bars show categories below the threshold (n=35). The dashed horizontal line marks the 25th percentile threshold (0.554%) used for filtering. The log scale allows visualisation of both dominant and rare categories spanning four orders of magnitude. b) Cumulative frequency distribution of spatial categories ranked by abundance. The top 17 categories (blue) capture 97.7% of the domain, validating the filtering approach for the following analyses.

## 3.2 Inter-model agreement assessment



Principal Component Analysis (PCA) offers a framework for multi-model agreement evaluation, by decomposing the temporal variability of multiple model time series into orthogonal components that distinguish common signals from model-specific divergences (Smith et al., 2012). Similarly, inter-model correlations of the seasonal variations provide insights into when and why CPM members fluctuate throughout the year.

# 3.2.1 Principal Component Analysis for inter-model agreement

Principal Component Analysis (PCA) has been extensively employed in atmospheric sciences to identify coherent spatio-temporal structures and modes of variability within Earth system datasets (Schmidt et al., 2019). This methodology focuses on identifying consistent signals and systematic divergences among models rather than making an absolute validation against observations (Bellucci et al., 2015). This multivariate approach addresses limitations of correlation analyses, which assess only pairwise relationships rather than the complete inter-model interaction structure. High inter-model agreement, measured through PCA loadings, may serve as a proxy for model reliability (Smith et al., 2012; Yan and Tang, 2013), as independent models with strong consensus indicate higher confidence in the representation of the underlying processes and, consequently, greater reliability of model outputs.





Following the mathematical framework presented in Jolliffe (2002), we treat each model as a variable in the analysis, enabling simultaneous quantification of consensus signals and divergences across the ensemble. We transform the three-dimensional space defined by the CPM simulations (ETH, CNRM, CMCC) into three principal components that capture the complete variance structure of inter-model relationships. At each grid point, the three time series represent the same meteo-rological variable (wind speed at 100 m) but simulated by different models under identical atmospheric conditions, enabling PCA to decompose the pure inter-model covariance structure that reveals systematic patterns in how models co-vary. The first principal component represents shared variance (consensus), while subsequent components capture structured disagreements between specific model combinations.

PCA was applied to daily wind gust time series derived from hourly 100 m wind speed data (2000-2009) simulated by each of the three CPMs (Section 3.1.5). Daily wind gusts are calculated as the maximum hourly wind speed for each day. To address the right-skewed distribution typical of wind data while ensuring mathematical validity for zero wind speeds, wind gusts are transformed using  $WG_{log} = log(1+WG)$ , where WG represents the original wind gust values in m s<sup>-1</sup>. Each time series was then standardised (by subtracting the mean and dividing by the standard deviation) to ensure equal contribution to the variance structure.

The loadings represent linear coefficients that quantify how each CPM contributes to each component, where  $PC_1 = (loading_{\rm ETH} \times {\rm ETH}) + (loading_{\rm CNRM} \times {\rm CNRM}) + (loading_{\rm CMCC} \times {\rm CMCC})$  (Jolliffe, 2002). PC1 loadings indicate each model's contribution to the consensus signal, with balanced positive loadings ( $\approx \frac{1}{\sqrt{3}} \approx 0.577$ ) indicating equal model contribution to the shared variance. PC2 and PC3 loadings reveal which models drive primary and secondary divergence patterns, respectively, with loading magnitude and sign indicating the direction and strength of each model's departure from the consensus. This loading-based analysis, applied across the 100 sampled points within each spatial category, enables systematic characterisation of how inter-model agreement varies across different spatial categories.

PCA shows the temporal concordance among the model simulations at each spatial location. The use of this method, in conjunction with stratified sampling across spatial categories, can provide an identification of locations where models show strong consensus versus areas where their differences reflect fundamental uncertainties in representing wind speed patterns. While this approach cannot establish absolute accuracy, it provides insights into the robustness of simulated wind patterns and identifies systematic behaviours or regions where models converge or diverge in their agreement, potentially revealing whether model uncertainties are surface-dependent or arise from fundamental formulation differences that persist regardless of geographic location, thus informing confidence levels in the simulations.

## 3.2.2 Seasonal agreement

While PCA analyses general patterns in the time series, direct monthly correlations reveal seasonal changes in agreement or identify which specific model pairs differ during certain months due to different atmospheric processes, such as winter storms or summer convection.

Seasonal patterns are relevant for wind energy applications, and assessing the agreement of the models on a seasonal basis may reveal periods of higher model uncertainty that could affect the reliability of extreme wind projections, especially dur-





ing months when convective processes dominate wind generation mechanisms. To evaluate seasonal patterns of inter-model agreement and identify temporal dependencies in CPM performance for extreme wind events, we conducted a monthly correlation analysis of wind speed maxima across the 10-year study period. This approach aims to capture how models consistently reproduce monthly wind extremes across inter-annual variations, providing insights into the temporal alignment of CPM performance across diverse European geographical and climatic contexts. Moreover, this analysis complements the PCA approach by providing direct quantification of pairwise model agreement at a monthly resolution, enabling identification of seasonal periods when model consensus is strongest or weakest.

For each spatial category and calendar month, monthly maximum wind speeds were extracted from hourly time series and used to compute Pearson correlations between model pairs (ETH-CNRM, ETH-CMCC, CNRM-CMCC) throughout the decadal period. Then, point-wise correlations were spatially averaged within each category to obtain representative monthly agreement profiles.

## 3.3 SMEV approach for wind speeds

The Simplified Metastatistical Extreme Value (SMEV) approach, proposed by Marra et al. (2019), relies on a different set of assumptions with respect to classical extreme value theory, which enables one to use the entire set of independent events, termed "ordinary events" instead of annual maxima or peaks over threshold only.

Namely, SMEV assumes that the class of the parent distribution describing the ordinary events is known. It then estimates the parameters of this distribution from all the available independent observations and explicitly accounts for their occurrence frequency (i.e., average annual number of ordinary events) instead of assuming this is infinitely high. Therefore, SMEV accounts for both the intensity and annual frequency of occurrence of the events, thereby providing a more direct physical interpretation of the atmospheric processes generating extreme winds (Marra et al., 2019). For the case of wind speed, the Weibull distribution represents a natural choice for the parent distribution, as supported by numerous previous studies (Cook et al., 2003; Cook, 2014; Harris and Cook, 2014; Tuller and Brett, 1984; Tye et al., 2014; Hennessey Jr, 1977; Palutikof et al., 1999).

## 3.3.1 Ordinary events identification for wind speeds

Ordinary events were identified through an iterative peak selection algorithm that identifies independent local maxima within the time series through a decorrelation-based separation. The largest value in the series is extracted as an ordinary event, and all the preceding and subsequent values within a temporal window defined by the temporal correlation function, including the identified maximum, are set to zero. The procedure is then iterated until the entire time series is explored.

The decorrelation time for the wind speed time series was calculated using the area-under-the-curve method (Kristensen et al., 2002; Larsén and Mann, 2006) applied to the autocorrelation function as in Eq. 1:

$$\tau_d = \int_0^{L_{\text{max}}} R(\ell) \, d\ell \tag{1}$$





where  $R(\ell)$  is the autocorrelation function at lag  $\ell$ , and  $L_{\rm max}=200$  hours. The autocorrelation was computed at 1-hour intervals from 0 to 200 hours to accurately estimate the decorrelation time, which defines the temporal separation required for identifying independent events in the extreme value analysis.

## 3.3.2 Weibull distribution to describe the tail of the ordinary events

Following the theoretical foundation established by Harris and Cook (2014) and Cook et al. (2003) regarding the appropriateness of Weibull distributions for parent wind speed populations, we use a two-parameter Weibull distribution to model the independent wind peaks. The Weibull cumulative distribution function  $F(x) = 1 - \exp[-(x/\lambda)^{\kappa}]$ , where x represents wind speed values,  $\lambda$  and  $\kappa$  represent scale and shape parameters, respectively, was fitted using least-squares linear regression in Weibull-transformed coordinates, following the methodology outlined by Marani and Ignaccolo (2015), whose codes are available from Marra et al. (2019). To address potential biases arising from low-intensity events that may not represent the upper tail characteristics, we follow Marra et al. (2023) to implement a left-censoring in which the upper portion of the distribution of the ordinary events is treated as exactly known in the estimation, while the values below the threshold are treated as non-exceedances. It follows that left-censoring is not equivalent to threshold exceedance approaches such as the peak over threshold of extreme value theory (in which the values below the threshold are thrown away). Based on the Marra et al. (2023) approach, we selected the top 10% of ordinary events (90th percentile threshold), balancing the need for sufficient sample size while ensuring representative characterisation of the upper tail that governs extreme value behaviour. Moreover, this approach allows for assessing the adequacy of the Weibull distribution through Weibull probability plots, with coefficients of determination  $(R^2)$  exceeding 0.9 for all CPM ordinary events in different locations.

## **3.3.3** Estimation of extreme wind return levels

Once ordinary events were identified and Weibull parameters estimated, extreme wind speeds for specified return periods were calculated using the SMEV formulation. The SMEV approach estimates return levels through the relationship SMEV $(x) = [F(x; \theta)]^n$ , where  $F(x; \theta)$  represents the fitted Weibull distribution with parameters  $\theta = (\lambda, \kappa)$  and n is the mean annual frequency of ordinary events (Marra et al., 2019). For each spatial category and model combination, return levels corresponding to 50-year return periods were computed using the quantile expression (Eq. 2):

$$U_T = \lambda \left[ -\ln \left( 1 - \left( 1 - \frac{1}{T} \right)^{1/n} \right) \right]^{1/\kappa} \tag{2}$$

where  $U_T$  represents the extreme wind speed for return period T, and the inner term (1-1/T) defines the annual non-310 exceedance probability. The mean annual frequency n was calculated as the average number of ordinary events per year across the 10-year simulation periods. The estimation procedure was applied independently to each of the three CPM datasets (ETH, CNRM, CMCC). All computations were performed at the individual grid point level before aggregation to spatial category

statistics, preserving the full spatial variability information within each category. The codes to test the SMEV assumptions and apply it are freely available from Marra (2020) and Marra (2022).

A bootstrap resampling was used to derive 95% confidence intervals for ensemble mean return levels ( $U_{50}$ ) (Tibshirani and Efron, 1993) for each disaggregated layer. These are calculated by pooling all points of categories within each layer level. To compute the confidence intervals, we first calculated the ensemble mean return levels by averaging the three CPM estimates at each of the 100 sampled points. Then, we derived bootstrap confidence intervals using 1000 resampling iterations at each point.

#### 320 4 Results





#### 4.1 Inter-model differences in wind extremes

We first quantify baseline differences in extreme wind representation among CPMs by analysing the spatial distribution of mean and coefficient of variation (CV, standard deviation divided by the mean) of annual maximum wind speed. These results are shown in Fig. 3, which reveals preliminary differences among the three CPMs. ETH exhibits systematically higher wind speeds across the inland domain, with average annual maxima ranging from 15-20 m s<sup>-1</sup> in low-lying areas to over 35 m s<sup>-1</sup> in mountainous regions and coastal zones. CNRM produces intermediate values with a more moderate spatial gradient, while CMCC consistently generates the lowest annual maxima, particularly in complex terrain where values remain below 20 m s<sup>-1</sup>. All models seem to reproduce a clear orographic enhancement and land-sea contrast effects on the extreme wind speeds. On the other hand, the CV reveals generally low inter-annual variability across all models (CV 

**Figure 3.** Annual maximum wind speeds: 10-year average (a-c) and coefficient of variation (d-f) for each pixel in ETH, CNRM, and CMCC models, respectively. The white outlines mark areas with elevations above 1,000 metres above sea level.

The second principal component (PC2, 14.4% of variance) captures the primary divergence pattern in temporal covariance. ETH loadings are centred near zero, CNRM shows predominantly positive loadings, and CMCC negative loadings, indicating that CNRM and CMCC systematically diverge in opposite directions from the ensemble mean; i.e., when CNRM predicts higher wind speeds, CMCC predicts lower speeds, with ETH falling between them. These patterns of variability are consistent with the spatial patterns observed in Fig. 3 d-f.

The third principal component (PC3, accounting for 11.4% of the variance) reveals a secondary, and independent, divergence pattern, in which ETH exhibits negative loadings. At the same time, both CNRM and CMCC show positive loadings with greater variability across categories. This pattern extends almost uniformly across all spatial categories, with the most pronounced differences occurring in the arid climates category.

Overall, the loading patterns remain remarkably stable across different spatial categories for PC1, PC2, and PC3, suggesting that inter-model relationships are consistent regardless of surface characteristics.

## 4.2.2 Seasonal variability


Figure 5 reveals distinct seasonal patterns in the inter-model correlations of monthly maximum wind speeds. The extended winter period, from October to March, shows consistently high correlations. The values usually range from 0.6 to 0.8 in most spatial categories, with January and March showing the strongest inter-model agreement. The pair of CNRM-CMCC models consistently shows the lowest correlations in all months, particularly during the winter, consistent with the divergence patterns

**Figure 4.** Principal components (PC) loadings distribution across spatial categories. a) PC 1, b) PC2, c) PC3. Points represent mean loadings, and error bars show the interquartile range (IQR) across the 100 sampled points within each spatial category.








in Fig. 3. The summer months (June through September) exhibit markedly reduced correlations, with July showing the most severe reduction, where correlations drop to 0.1-0.4 for all categories.

The seasonal correlation results confirm the robust inter-model hierarchy: CNRM-CMCC consistently ranks lowest, while ETH-CMCC and ETH-CNRM are roughly equal, regardless of the spatial category. Instead of surface-specific variations driving the observed correlation, the ordering of model pairs remains stable across diverse surface types, with spatial correlation variations typically being modest ( $\sim$ 0.2 range), implying consistent differences between the model pairs. However, there are exceptions in months like October and February, where spatial correlation exhibits broader spread ( $\sim$ 0.4), indicating that seasonal forcing can amplify surface dependencies. Additionally, transitional months (April-May, October-November) show intermediate correlation levels between winter peaks and summer troughs, with increased inter-category correlation spread.

#### 4.3 Extreme wind return levels

Figure 6 presents the distribution of 50-year wind return level ( $U_{50}$ ) estimates. Figure 6a-c illustrates the inherent intra-model variability across spatial categories (disaggregated by each layer). In d-f, the ensemble return levels (boxplot, average extreme wind estimates of each single CPM member) are compared with the sampling uncertainty (Bootstrap 95% confidence interval shown as grey bands). Sample sizes vary substantially across categories (Fig. 6g-i), with temperate and cold climates providing the largest samples (700 and 600 points, respectively), while arid and polar climates contribute smaller samples (100 and 200 points). Roughness category  $R_5$  dominates the domain with 900 points, while water surfaces ( $R_1$ ) and  $R_2$  provide limited and null representation. The topographic distribution is more balanced, though  $T_1$  contributes the largest sample (700 points) compared to  $T_2$ – $T_4$  (200–400 points each).

The differences among models remain uniform across spatial categories (Fig. 6a-c). ETH consistently produces the highest extreme winds across all spatial categories, showing the most consistent estimates within categories. Conversely, CMCC generates the lowest estimates, with differences of approximately 10–15 m s<sup>-1</sup> in the central distribution among the models and has greater internal variability. CNRM has intermediate behaviour with moderate dispersion within categories, which is closer to ETH patterns.

The ensemble mean  $U_{50}$  return levels within their bootstrap confidence intervals (Fig. 6d-f) reveal differences in extreme wind speed across spatial categories. A clear progression can be noticed, with tundra (Td) and cold (Co) climates having higher ensemble means ( $\sim$ 28 m s<sup>-1</sup>) compared to arid (Ar) climates ( $\sim$ 23 m s<sup>-1</sup>). Water surfaces  $(R_1)$  show the highest inter-model consistency, while terrestrial roughness categories  $(R_3-R_5)$  exhibit greater model disagreement despite similar ensemble means. Topographic categories demonstrate both similar ensemble means and comparable inter-model variability, with all terrestrial categories  $(T_1-T_4)$  yielding consistent extreme wind estimates of 25- 27 m s<sup>-1</sup>.

The ensemble approach captures the central tendency across the three CPMs, providing more balanced extreme wind estimates than using individual models. The bootstrap confidence intervals confirm the statistical robustness of the ensemble means across all spatial categories. Moreover, it shows consistent inter-model differences for all surface types. This indicates that the estimation uncertainty comes mainly from model formulation rather than surface-dependent processes.

Figure 5. Monthly maxima wind speed correlation per spatial category and CPMs couple.

## 5 Discussion


## 5.1 Implications of inter-model differences

CPMs explicitly resolve convective processes, allowing each model to develop its own mesoscale meteorology, including convective systems, localised circulations, and fine-scale momentum transport processes (Coppola et al., 2020). One of the first findings is the systematic hierarchy in the annual maxima magnitudes observed in Figure 3, with ETH > CNRM > CMCC,



**Figure 6.** Wind return levels with 50 years return period estimated per model and spatial category (a-c). Distribution of the ensemble average 50 years of extreme wind speeds for each spatial category with 95% confidence intervals from bootstrapped samples (d-f). Number of points used per spatial category in the estimation of the extreme winds (g-i).

suggesting that each model's treatment of these explicitly resolved processes leads to systematically different extreme wind speeds at 100 m height.

Figure 3 provides information on the physical processes driving this model hierarchy. On one hand, continental areas, particularly those with mountainous terrain, exhibit the largest inter-model spreads in annual maxima, reflecting the complexity of representing orographic flows, channelisation effects, and topographically induced turbulence, where small-scale processes strongly influence wind field development. These differences in Fig. 3a-c can exceed 15-20 m s<sup>-1</sup> over mountainous terrain, and are important because they could significantly alter turbine design loads and compromise structural safety analyses, making the selection of a single CPM a critical source of uncertainty. On the other hand, marine environments show reduced intermodel variability, in line with more spatially uniform surface characteristics that reduce sensitivity to differences in surface-atmosphere coupling approaches. Moreover, the coefficient of variation patterns in Fig. 3d-f reveal that CNRM exhibits greater interannual variability in extreme wind speed, particularly in complex terrain, indicating higher sensitivity to meteorological forcing variations compared to ETH and CMCC. These findings challenge the reliability of isolated CPM applications for


extreme wind estimation and demonstrate that inter-model differences require an evaluation to distinguish robust physical signals from model-specific artefacts. Without an assessment of inter-model divergences, extreme wind estimates lack the statistical foundation necessary for engineering applications where structural integrity depends on accurate load projections.

## 5.2 Physical interpretation of model agreement patterns

The PCA (Fig. 4) reveals insights into the CPM representation of extreme wind processes across different surface characteristics, preserving the systematic hierarchy ETH > CNRM > CMCC detected in the previous analysis. The dominant mode (PC1, accounting for 74.2% of variance) represents a common climatological signal, with consistent loading patterns across spatial categories, suggesting that this fundamental signal is robust regardless of surface characteristics. This likely reflects their shared response to large-scale synoptic forcing from ERA-Interim boundary conditions, such as the passage of extratropical cyclones and frontal systems that dominate extreme wind generation across central Europe.

The disparities shown by PC2 and PC3 demonstrate how the models vary in their extreme wind simulations. PC2 identifies the main axis of disagreement among models, showing how CNRM and CMCC diverge in opposite directions while ETH stays close to the overall consensus. On the other hand, PC3 reveals a secondary divergence in which ETH separates from the CNRM-CMCC duo, as evidenced by persistent sign patterns in the loading distributions across all spatial categories (Fig. 4d). Because PC2 and PC3 are orthogonal, these divergences are statistically independent. This implies that the systematic differences among models originate from various uncorrelated sources within the CPMs formulations, rather than being driven by a single dominant factor. These independent divergence patterns likely refer to differences in how model physics are implemented, for example, PC2 might capture variations in convective parameterisation schemes and vertical momentum transport, while PC3 could reflect differences in boundary layer mixing and surface flux calculations; both crucial areas where CPMs settings can differ significantly (Coppola et al., 2020; Fosser et al., 2024).

Furthermore, the consistent loadings across diverse surface types suggest that each CPM maintains its characteristic be-430 haviour independent of local surface conditions, pointing to systematic (not random or specific) differences in model physics as the primary drivers of inter-model uncertainty. These results establish that ensemble approaches are necessary because single-model representations of wind fields could provide complementary information.

# 5.3 Seasonal and spatial dependence in model agreement

A pronounced seasonal cycle in inter-model correlations of the monthly maxima is observed (Fig. 5), providing clear evidence for the physical mechanisms controlling extreme wind generation and revealing fundamental differences in how CPMs represent diverse meteorological regimes. Winter months exhibit consistently higher correlations across most spatial categories, reflecting the dominance of extratropical cyclones and large-scale synoptic systems in generating extreme winds. This synoptic-scale inter-model agreement is the result of the ERA-Interim boundary forcing, which provides the common meteorological boundaries that all three models represent similarly. In contrast, correlations are lower during summer months, especially July, reflecting a shift in the physical processes generating extreme winds. During summer, large-scale synoptic systems are weakened, and extreme wind events become increasingly driven by localised convective processes, thermal circu-




lations, and mesoscale phenomena. In the summer, in the absence of strong, large-scale synoptic forcing, each CPM internally develops its own mesoscale meteorology, including convective systems, land-sea breezes, and orographically influenced flows. This independent representation of sub-grid phenomena explains why model agreement decreases when these internally driven processes dominate the generation of extreme winds.

The spatial behaviour of seasonal correlation patterns reveals that CNRM-CMCC consistently exhibits the lowest correlations across all months and surface categories. At the same time, ETH-CMCC and ETH-CNRM show comparable and alternating correlation levels throughout the seasonal cycle (Fig. 5). This pattern is consistent with the PC2 results, where CNRM and CMCC systematically diverged in opposite directions while ETH remained near the consensus. Rather than surface-specific variations driving inter-model agreement, the almost consistent correlation patterns across diverse surface types reinforce that systematic model differences dominate over surface-atmosphere interaction effects. However, while spatial variations in correlations are typically modest (~0.2 range), certain months such as October and February show substantially larger spatial dependency (~0.4 range), suggesting that specific seasonal conditions may amplify surface-related effects on inter-model agreement.

Future climate projections of extreme winds in convectively active regimes will likely carry greater uncertainty and would benefit significantly from multi-model ensemble approaches to capture the range of possible mesoscale responses to changing thermodynamic conditions. In addition, particular care should be taken regarding the possible changes in the seasonality of large-scale systems, such as extra-tropical cyclones.

## 5.4 Estimation of extreme wind return levels

The application of SMEV to estimate extreme wind return levels from CPMs is a novel contribution of this study. The systematic differences among individual models and the ensemble's central tendency provide a benchmark for future CPM evaluation studies, while the spatial category approach enables targeted assessment of model performance across different surface types relevant to wind energy development.

The advantage of using SMEV instead of a traditional extreme value approach comes from the short duration of CPM simulations. While traditional extreme value analysis depends on annual maximum values, which are just 10 values from the length of the simulations, or a still limited number of peak values above a very high threshold, SMEV uses the complete set of independent events. This provides abundant information for estimating extremes from the available short time series, thereby addressing this critical limitation of CPM simulations. This is particularly important for wind energy, where design standards require estimates for 50-year return periods that typically need to be based on limited simulation periods.

The assumption of Weibull tails for the distribution of the wind ordinary events, while well-supported for wind data (Harris and Cook, 2014), may not fully capture wind field complexity in heterogeneous terrain where multiple meteorological mechanisms contribute to extreme generation (Harris and Cook, 2014). However, this potential limitation is mitigated in the present study through the use of left-censoring, by which the potential impact of heterogeneous wind processes with different typical intensities is reduced.




Using ensemble approaches offers clear benefits over relying on single-model assessments for extreme wind evaluations. The individual models in Fig. 6a-c follow the same consistent hierarchy (ETH > CNRM > CMCC) across all spatial categories, as evidenced in previous results. This reinforces the idea that the way models are formulated leads to differences between them, rather than specific surface effects. In this context, CMCC shows greater internal variability, as indicated by its wider interquartile ranges, compared to ETH and CNRM. This suggests varying levels of dispersion within categories that warrant further investigation.

Some spatial categories, like Ar-arid climates and  $R_1$ -water surfaces, show less variability among the models. This probably reflects the more stable and limited range of meteorological conditions typical of these environments. Additionally, the ensemble approach in Fig. 6d-f gives more balanced estimates by averaging these systematic model differences and varying internal sensitivities. This helps avoid the potential bias that could arise from using a single model. These patterns confirm that while the model hierarchy is consistent, individual models respond differently to sub-grid processes, highlighting the importance of ensemble approaches in capturing the full spectrum of physical process representations.

The bootstrap confidence intervals in Fig. 6d-f provide strong statistical uncertainty bounds that are much narrower than the spatial variability seen within ensemble estimates across each category. This offers wind energy practitioners a more reliable range for design purposes, compared to the broader variability found in the full ensemble data.

## 490 5.5 Implications for wind energy applications

The consistent differences in how extreme winds are represented among CPMs and their spatial patterns offer insights for wind energy applications that go beyond traditional meteorological analysis. The variability of seasonal agreement among models provides a framework for assessing uncertainty and enables wind industry experts to adjust their confidence levels based on the season. For example, one could consider applying higher uncertainty margins during the summer months, when convective processes are more prevalent and inter-model correlations tend to decrease. In contrast, more confidence can be placed in model consensus during winter periods, when synoptic processes dominate and correlations increase. The uniformity of model differences across all surface types indicates that ensemble approaches are essential, regardless of the project's location.

In terms of wind resource assessment, this systematic model hierarchy allows for risk-based approaches. For critical infrastructure design, conservative estimates using ETH can be employed, while ensemble estimates offer balanced projections for standard commercial development. Given the consistent differences among models across all spatial categories, adopting ensemble approaches should be standard practice for evaluating extreme winds, irrespective of surface type or geographic location. Insights into the systematic behaviours of models identified in this study can assist wind energy professionals in understanding the inherent uncertainties in CPM-based projections and in implementing appropriate ensemble strategies that align with their risk tolerance and application needs.

## 505 5.6 Limitations of this study

A fundamental limitation of this study is the lack of observational datasets with sufficient spatial coverage, temporal extent, open accessibility, and consistent measurement heights to enable direct validation of CPM simulations. This observational gap






limits our ability to validate the physical realism of the observed model hierarchy and to assess whether the  $U_{50}$  estimates align with real-world extremes across spatial categories. Additionally, the relatively short 10-year simulation period, while sufficient for obtaining reasonable estimates with SMEV, represents a temporal limitation that future studies could address by incorporating longer simulations to better capture interannual and decadal variability in extreme wind patterns. Also, an increased number of ensemble members is needed for a more robust assessment of model uncertainty.

Future validation efforts would greatly benefit from extensive observational networks that provide long-term wind measurements at hub height across diverse surface types. Specific validation campaigns in representative spatial categories could be helpful to assess absolute model performance and bias characteristics on the typical wind conditions, but would hardly provide actionable information about the extreme return levels needed for turbine design. To this end, priority should be given to the already available observational datasets, which should be made available to the community as open data.

#### 6 Conclusions

This is the first application of CORDEX-FPS CPMs for assessing extreme wind at turbine heights. We first investigate whether there are discrepancies among the models and how well the models agree in representing winds at 100 m. We demonstrate substantial inter-model differences in extreme wind representation, with systematic variations exceeding  $15-20 \text{ m s}^{-1}$  in annual maxima values in complex terrain despite identical geographic domains and atmospheric forcing. Furthermore, we provide a framework based on model inter-comparison for evaluating the CPM simulations of extreme wind for different spatial categories in the absence of long-term observational data. We then combine these simulations with the Simplified Metastatistical Extreme Value approach (SMEV) to assess extreme wind return levels at turbine heights.

We organise the analyses into spatial categories based on climate, terrain roughness, and topographic complexity to isolate the impact of these features and check if they had an effect on the divergences of CPM simulations. This approach provides category-specific estimates of extreme wind, along with category-specific confidence intervals, which can offer practical information for wind energy development and turbine design in various physiographic settings.

We applied the Principal Component Analysis (PCA) to understand the relative agreement among models, and show that 74.2% of variance represents a strong common climatological signal while the rest of the variance captures systematic and independent divergences among CPMs that remain uniform across all spatial categories. This consistent nature of the differences across all surface types indicates that model formulations, rather than the surface characteristics, are what mainly drive the inter-model uncertainty in extreme wind projections. This highlights the need for ensemble approaches, regardless of geographic location. Moreover, the significant seasonal variation in model agreement (with higher correlations in winter and lower agreement in summer) reflects the different effects of synoptic and convective processes in creating extreme winds, which also offers valuable insight for quantifying seasonal uncertainty in wind energy applications.

We find a consistent hierarchy in the simulated extreme wind speeds across all spatial categories, which is useful for designing risk strategies in wind resource assessment. Furthermore, the application of SMEV to short CPM simulations extends its

https://doi.org/10.5194/wes-2025-172 Preprint. Discussion started: 30 October 2025

© Author(s) 2025. CC BY 4.0 License.




WIND
ENERGY
SCIENCE
DISCUSSIONS

applicability for obtaining extreme wind estimates from limited-duration datasets, establishing its viability for extreme wind

assessment in the wind energy context.

Overall, our work establishes a quantitative framework for evaluating the performance of CPMs in estimating extreme winds and could serve as a basis for future work in wind resource assessment, turbine site placement, and grid integration planning across various surface conditions and climates. Given that individual CPM members hold differences in representing extreme

wind patterns, approaches based on an ensemble of models are fundamental.

CPM projections are already available for near- and far-future scenarios. Combining these projections with the framework presented here could be a viable way to conduct future climate change impact assessments for the wind energy sector.

Code availability. The scripts for the formal analyses carried out in this work are available at Correa-Sánchez (2025). Codes to run the

SMEV model and test its applicability are available from Marra et al. (2019) and Marra (2022).

Data availability. The CPM data used in the study cannot be shared by the authors, but they are available at the *Metagrid* user interface of the infrastructure software Earth System Grid Federation (ESGF) at https://esgf-metagrid.cloud.dkrz.de/search for the

CORDEX-FPS CONV project.

*Author contributions.* NCS, Data curation, Formal analysis, Software, Visualisation, Writing (original draft preparation); NCS, XL, FM Conceptualisation, Methodology, Investigation; XL, FM Supervision; MB Funding acquisition; NCS, XL, ED, MB, FM Writing (review and editing).

Competing interests. The authors declare that they have no conflict of interest.

Acknowledgements. This study was partially supported by the CARIPARO Foundation through the Excellence Grant 2021 to the "Resilience" Project. FM was supported by the "The Geosciences for Sustainable Development" project (Budget Ministero dell'Università e della Ricerca–Dipartimenti di Eccellenza 2023–2027 C93C23002690001). ED was supported by the RETURN Extended Partnership and received funding from the European Union Next-Generation EU (National Recovery and Resilience Plan – NRRP, Mission 4, Component 2, Investment 1.3 – D.D. 1243 2/8/2022, PE0000005). XL acknowledges support from Horizon Europe DTWO project (101146689). We thank Marianna Adinolfi for providing the roughness layer used in the COSMO model for this study.

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
