# Peer review of "Assessing Inter-Model Agreement in Convection-Permitting Simulations of Extreme Winds for Wind Energy Applications"

_Wind Energy Science, 2025_

## Referee Comment (RC1)

**General feedback**

This is a very interesting study that I find to be complete, focused and clearly written. It brings many clear and novel results. The methods are explained well, the chosen figures are clear and well-chosen and there is a very good discussion about implications and limitations of the work. Still, I have taken my time to formulate several general and specific comments that I believe can improve the manuscript.

**General comments**

- 1. As you have the wind speed timeseries for a 10-year period available, I believe it would be a valuable addition to, at least briefly, discuss how the models compare in terms of the full wind speed distribution, not just the extremes. It would be a very interesting finding if the overall distributions (means, variability, histogram overlap) match well, as you find strong inter-model differences for the extremes. In other words, good performance on the overall distribution might not imply that you can just use any single model for the extremes.
- 2. There is still quite a lot of variability in the U50 estimates within individual spatial categories. Of course, the sites within a category might still be different in several ways, which can explain this. However, could it also be that over the 10-year period, some of these sites encountered a much more severe convective storm than other sites? Or more generally, that the intra-category, point-specific sets of ordinary events differ due to different sampling of meteorological events in these 10 years? I can imagine that this would lead to other U50 estimates. This would have several implications for wind energy: Using the site-specific U50 estimate, or even the ensemble mean U50, might not be a "safe" choice but one might be better off considering the grid cell of that category that was most "unlucky" during this 10-year period due to stochasticity. If it is correct that meteorological stochasticity over the 10 years plays an important role in the spread, please explain this to reader, e.g. by integrating into section 5.4 and 5.5. If, on the other hand, you believe that the spread is mainly due to varying site surface conditions, I would recommend to add a small section in the discussion that it is crucial to perform a site-specific assessment for extreme winds when a new project is developed somewhere. Perhaps both of these perspectives deserve a space?
- 3. The SMEV analysis you present is very interesting. However, I think it would be valuable to the work to add some sensitivity testing to illustrate the robustness of the method. For example, how sensitive is the method to the chosen parent distribution (I believe there are other distributions used to model extreme wind speeds)? Or how sensitive is it to the threshold chosen for the left-censoring? Perhaps you can also elaborate somewhere how your bootstrapping supports the robustness of this method.

**Specific comments**

**Line 8-9**: after reading a bit further, I realized that this sentence is quite confusing for several reasons. You summarize your complete methodological package into one short sentence, which I think needs elaboration. I would rework this to make things more clear. Three reasons why it is unclear:

- The term 'surface-based spatial categorisation' is vague when no additional information is added. "Surface" on itself can refer to many things (e.g. also surface winds) and only later you explain that you classify into terrain, roughness and climate zones, I think you can be a bit more elaborate and specific already in the abstract.
- The way the sentence is written now, the reader might think you used PCA for the spatial categorisation.
- You mention PCA, a very widely used and general method, but do not explain here what you use it for in your work, so also here I advise to be a bit more specific and elaborate.

**Line 26-27**: This sentence is unclear to me. It could be because regional variability is mentioned twice, but in different ways, so I think the sentence can be more concise.

**Line 31: A few comments**

- Please distinguish between GCM and RCM resolutions, as GCMs have a nominal horizontal grid increment of >=100 km, not 10 km.
- Using a term like "horizontal grid spacing" might also be a better idea than resolution, because the effective resolution is many times lower than the grid spacing. You could also clarify with a sentence that you are referring to grid spacing, not effective resolution.
- Please be more specific with what you mean with "general" wind patterns. GCMs only capture synoptic and larger scales. RCMs (dx > 10 km )also capture some mesoscales. Perhaps it's better to describe which scales both of them do not capture, that the CORDEX FPS runs do capture.
- Also, I recommend changing "cannot capture convective motions" to "cannot explicitly resolve..."

**Line 55-64:** This entire section is about methods you will not use eventually. Therefore, I would suggest to shorten it a bit and also to merge it with the section where you introduce SMEV, the method that you will be using. More generally, I would check where else in the introduction you could make the sequence more logical, because I feel like you switch back and forth between surface characteristics, inter-model comparison and EV estimation quite a bit, which is a bit difficult for the reader.

**Line 68-69**: This sentence is a bit confusing for me, as I don't see how the parts are connected. With addressing the limitations of observations with an inter-model comparison, do you mean that models provide spatial fields that observations cannot and therefore capture the spatial variability of signals? That would be valid, but I don't understand why the part of performing a PCA-based inter-model comparison is merged into that sentence. Perhaps elaborate and/or split it up into more simple chunks.

**Line 91-96**: After reading the manuscript I went back to these objectives. I think that it would help to be a bit more explicit that objective 1 is a full-domain analysis based on spatial maps. And that objective 3 and 4 build on objective 2 to perform analyses for the different categories, because the link is not very clear the way it is written now.

Line 113-116: Model physics (e.g. convection scheme) and stochastic variability are indeed an important part of why the CPMs might differ, but I think that the following elements can also be important:

- Model numerics: e.g. the number of vertical levels used and/or the amount of numerical diffusion applied, .... Could you add something about it to the text?
- Simulation protocol: continuous vs. chunked simulations, temporal spin-up. Perhaps the models had the same settings here? Could you add some notes on this?
- Also, next to the convection scheme, also the parametrization of the planetary boundary layer (PBL) is important in shaping low-level wind variability – the CPMs might be using different ones. Please mention this too. It would be helpful to add into the table the convection & PBL schemes used by the different models. Or at least to give a general idea of how they differ between models?

**Fig 2:** If the 25th percentile of the relative frequency of the categories is used, How can 35 out of 51 categories have relative frequency below this value? That does not agree with the definition of percentiles. Please clarify in the text so that the reader can better understand this.

**Line 208:** I think "temporal variability" should be changed to "temporal co-variability", because you do not perform the PCA on the individual time series.

**Line 210:** What do you mean with "fluctuate"? If you mean "why models differ", then I would not use the word fluctuate. If you mean some form of fluctuation, please be more specific about what you mean.

**Line 224**: "under identical atmospheric conditions" – I don't think this is correct because the model itself creates the atmospheric conditions in the domain. What I think you can say is "under identical atmospheric boundary conditions".

**Section 3.2.2**: I agree that this seasonal correlation analysis is a nice addition to the PCA. Yet, you perform this on the absolute monthly maximum – could it not be informative to also perform this analysis on, for example, the 99th percentile? I expect that the P99 wind speed value is also very relevant for wind energy planning and will be less "noisy" than the monthly maximum, sort of a "less anomalous, but still very extreme" wind speed. You could add this, or you could add a good argumentation about why the analysis of both is equivalent.

**Line 268**: "Ordinary events" you say this is the entire set of independent events, but it is not clear what these independent events means at this point. In section 3.3.1 it becomes clear that these ordinary events are extracted local maxima separated with your correlation window. Therefore, I would be more careful with introducing the term already here because it can be confusing to the reader. If you do introduce it here, please explain what it means.

**Line 271**: "instead of assuming this is infinitely high" – why would an algorithm assume that the frequency of occurrence of events is infinitely high? Can you please explain this part better please?

**Line 272**: When you say that it accounts for "intensity" do you mean that it takes into account the intensity distribution? Because "intensity" seems to refer to one specific intensity value. Please clarify this a bit more in the text.

**Section 3.3.2:** This comment links back to my last general comment. Normally, Weibull distributions are used to fit an entire wind speed distribution, but you already filter for local maxima and then fit a Weibull distribution. Even for full wind speed timeseries, Weibull distributions don't always work well for complex sites and I am also surprised that here it is applied to a maxima-filtered timeseries, with the "left-censoring" added to it. I think it is necessary to convince the reader (visually) that this fitting works. Could you either add a few supplementary plots or refer to some other work where this is clearly demonstrated?

**Line 315:** Can you be more explicit what you bootstrapped? Is it the ordinary event dataset for each model? Is it the different points in each category? This is currently not very clear from this small section. Please clarify in the text.

**Line 316**: "for each disaggregated layer" – I think this is the first time you use the term "layer" and it is unclear what you are referring to. Could you please clarify in the text?

**Figure 6**: Can you add a physical interpretation in the text on why the lowest roughness class, i.e. open water areas, is characterized by the lowest U50 wind speed estimates? Because low roughness corresponds to less frictional deceleration so in this sense you would expect higher extreme wind speeds. Are there perhaps less summer convective storms over water? (I don't know immediately myself).

**Conclusion**: I think it would be good to — very briefly — repeat some of the suggestions/limitations you mentioned here again. For example, the need for CPM evaluation for observations, measurement campaigns and the need to involve more CPMs. Also, I would recommend to use the past tense for summarizing the stuff that you did. For example "We first investigate..." → "First, we investigated...". General truths or findings can remain in present tense of course (e.g. "This highlights the need for ensemble approaches..."). I personally think this makes more sense, but you can also leave it like this if you don't agree.

---

## Author Comment (AC1)

**RESPONSES POINT BY POINT TO REFEREE N°1 OF WIND ENERGY SCIENCE DISCUSSIONS**

*Assessing Inter-Model Agreement in Convection-Permitting Simulations of Extreme Winds for Wind Energy Applications*

**Nathalia Correa-Sánchez, Xiaoli Guo Larsén, Eleonora Dallan, Marco Borga, and Francesco Marra**

Dear Referee No.1,

We thank you for your review work and the valuable comments, which helped to improve our paper. Our responses are reported in blue, and all the modified or new text is reported in *italics and red*. Line numbering refers to the original version of the paper that was available for the open discussion. Reviewer(referee) comments are numbered as RxGCy for general comments and RxSCy for specific comments, where x indicates the reviewer number (1 or 2) and y indicates the sequential comment number.

**General feedback**

This is a very interesting study that I find to be complete , focused and clearly written. It brings many clear and novel results. The methods are explained well, the chosen figures are clear and well chosen and there is a very good discussion about implications and limitations of the work. Still, I have taken my time to formulate several general and specific comments that I believe can improve the manuscript.

**General comments**

**R1GC1.** As you have the wind speed timeseries for a 10 year period available, I believe it would be a valuable addition to, at least briefly, discuss how the models compare in terms of the full wind speed distribution, not just the extremes. It would be a very interesting finding if the overall distributions (means, variability, histogram overlap ) match well, as you find strong inter-model differences for the extremes. In other words, good performance on the overall distribution might not imply that you can just use any single model for the extremes.

**R//:** We appreciate this suggestion. We have added supplementary analysis of the full wind speed distributions (Figure S1) in the annexes, including the probability density functions and the exceedance probability in log-log scale for the complete 10-year hourly time series.

[Figure]

**Figure S1.***Statistical distributions of hourly wind speed at 100 m for the full 10-year time series (2000-2009). (a) Probability density function showing the overall distribution with summary statistics. (b) Exceedance probability in log-log scale, highlighting tail behaviour and focusing on the extreme wind range (10-40 m/s) where the grey horizontal lines indicate the 95th, 99th, and 99.9th percentiles.*

This analysis reveals inter-model differences in bulk statistics. On the one hand, ETH and CNRM show similar means and medians, while CMCC exhibits systematically lower values, yet the models converge in representing extreme winds. This demonstrates that bulk distribution performance does not predict extreme-value representation, which addresses the reviewer's hypothesis by showing that models with different bulk behaviour can produce consistent extremes, thereby reinforcing the need for dedicated extreme-value analysis in wind engineering applications.

**Figure S1** shows that one cannot infer extreme-value reliability from bulk distribution statistics. For this reason, we have added the following text in line 322: *"Figure S1 shows that CPMs with different bulk statistics (panel a) can produce similar extreme wind behaviour (panel b, winds >25 m/s), confirming the need for dedicated extreme-value analysis in wind engineering. Given this, we first quantify baseline differences…"*

**R1GC2**.There is still quite a lot of variability in the U50 estimates within individual spatial categories. Of course, the sites within a category might still be different in several ways, which can explain this. However, could it also be that over the 10 year period, some of these sites encountered a much more severe convective storm than other sites ? Or more generally, that the intra category point specific sets of ordinary events differ due to different sampling of meteorological events in these 10 years. I can imagine that this would lead to other U50 estimates. This would have several implications for wind energy: Using the site specific U50 estimate, or even the ensemble mean U50 might not be a "safe" choice but one might be better off considering the grid cell of that category that was most "unlucky" during this 10 year period due to stochasticity. If it is correct that meteorological stochasticity over the 10 years plays an important role in the spread, please explain this to reader, e.g. by integrating into section 5.4 and 5.5 If, on the other hand, you believe that the spread is mainly due to varying site surface conditions, I would

recommend to add a small section in the discussion that it is crucial to perform a site specific assessment for extreme winds when a new project is developed somewhere. Perhaps both of these perspectives deserve a space.

**R//:** We thank the reviewer for bringing this discussion to the table. We recognise that both factors contribute to the observed variability. Overall, our message is that CPMs can provide robust climatological information at the regional/categorical scale (demonstrated by PCA: 74.2% common variance), and that the intra-category variability reflects:
1. Temporal sampling variability: during the 10-year range period, some grid cells had more severe meteorological events (e.g., intense convective storms) by stochastic chance, which generates different ordinary event sets and U50 estimates.
2. Spatial surface heterogeneity: fine-scale differences in local roughness, topographic features, and land-surface properties that exist even within our spatial categories. This is because  of the discretisation of continuous geophysical properties: roughness levels that span ranges (e.g., R3 are z_0: 0.03–0.3 m), topographic levels that aggregate varying terrain features, and climate types that group Köppen-Geiger subtypes.

In this sense, and following your suggestions, we have added the paragraph below the line 489 of section 5.4:
*"The intra-category variability in U_50 estimates shows the combined effects of meteorological sampling over the 10-year period and  fine-scale spatial heterogeneity. During these temporal windows, different sites may have varying extreme meteorological conditions, with some experiencing more severe convective storms or synoptic extremes randomly, leading to different sets of ordinary events and consequently varying U50 estimates. Additionally, local surface characteristics (fine-scale roughness, microtopography) vary within categories; although this heterogeneity is less than the heterogeneity among categories, it is physically reasonable because categories inevitably capture dominant patterns, not absolute uniformity. In this sense, both factors represent features of high-resolution climate  data rather than methodological limitations."*

And in line 505 of Section 5.5, we have addressed the discussion on the practical implications of this by  adding a new paragraph:
*"The intra-category variability of U_50 has practical implications for multi-scale wind resource assessment and long-term planning. Our CPM ensemble approach provides robust climatological information at the regional and categorical scales (3 km horizontal grid spacing), as evidenced by the strong common signal (74.2% of variance), which is valuable for regional resource mapping and preliminary site identification. However, when transitioning to site-specific turbine design, wind energy specialists require micrositing assessments at sub-grid scales (1-100 m) to capture fine-scale terrain features, local obstacles, and surface roughness variations not resolved by CPMs. This site-specific refinement follows established industry protocols and is necessary regardless of the regional climate data source employed."*

Furthermore, in line 518 of Section 5.6 on the limitations of the paper, we have explained more clearly the utility position of CPMs at the assessment scale.

*"The 10-year simulation period provides enough temporal coverage for SMEV application, as demonstrated by the strong inter-model consensus. However, this simulation period captures a finite sample of possible meteorological events, and some intra-category spread in U_50 estimates reflects this meteorological sampling variability. Future studies with longer simulation periods (20-30 years) would further refine the separation between systematic spatial patterns and temporal sampling effects."*

**R1GC3**.The SMEV analysis you present is very interesting. However, I think it would be valuable to the work to add some sensitivity testing to illustrate the robustness of the method. For example, how sensitive is the method to the chosen parent distribution (I believe there are other distributions used to model extreme wind speeds) Or how sensitive is it to the threshold chosen for the left censoring? Perhaps you can also elaborate somewhere how your bootstrapping supports the robustness of this method.

**R//:** We thank the reviewer for this suggestion.

Regarding the parent distribution, it is true that other valuable works such as Carta et al.(2009) provide an extensive review on this matter that includes, lognormal, generalized gamma, and three-parameter Weibull variants. Here, we choose the two-parameter Weibull distribution since it also has extensive empirical validation for wind speed modeling (Justus et al., 1978; Hennessey, 1977; Tuller and Brett, 1984; Cook et al., 2003; Carta et al., 2009; Harris and Cook, 2014).

It should be noted that we use left-censoring in our estimates, meaning that we can model accurately any distribution which is Weibull-right-tail-equivalent. We conducted Weibull probability plots over the ordinary events to confirm the appropriateness of this model for our data. To illustrate, we have added 12 randomly selected time series to display these Weibull plots in the annexes **(Figures S4-S6)** .

In addition to the referred literature supporting this choice, it is important to note that, while distributional choice affects absolute return levels, our comparison of the models is expected to be more reliable because: i) all CPMs are assessed under the same Weibull distribution, applying any potential distributional biases equally; and ii) the Weibull probability plots **(Figures S4-S6)** show that all three CPMs display similar upper-tail behavior, which supports using a common distributional assumption.

Now, regarding the threshold, following Marra et al. (2019, 2020, 2023), we use 90th percentile, which we confirmed with the Weibull plots, the new addition of the annexes. It should be noted, as detailed in Marra et al. 2023, that once the tail model is appropriate, there is virtually no sensitivity of the results to this threshold (any threshold larger than the one that isolates the Weibull tail will provide the same results - aside from estimation uncertainty).

Having this in mind, we modified the text to clarify on these aspectecs by adding in line 275: *"...is and Cook, 2014; Tuller and Brett, 1984; Tye et al., 2014; Hennessey Jr, 1977; Palutikof et al., 1999) and has been shown to effectively approximate distributions with theoretical foundations in atmospheric physics (Harris and Cook, 2014)…."*

In this sense, and also responding to **comment R1SC16**, we added in line 301: *"...ordinary events in different locations. To illustrate, Weibull probability plots from 12 randomly selected locations of each CPM are provided in the supplementary material (**Figures S4-S6**; **Table S1** shows their location and the spatial category to which they correspond). The linear relationship in ln-ln space validates the Weibull distribution assumption for the upper tail across diverse geographical and climatic conditions."*

[Figure]

**Figure S4.** *Weibull probability plots for ordinary events from the ETH model.*
*Each panel shows ordinary events (gray), the top 10 % data portion (in red), and the linear fit (black dashed line) of that top portion in randomly selected locations.*

[Figure]

**Figure S5.** *Weibull probability plots for ordinary events from the CNRM model.*

*Each panel shows ordinary events (gray), the top 10 % data portion (in red), and the linear fit (black dashed line) of that top portion in randomly selected locations.*

[Figure]

**Figure S6.** *Weibull probability plots for ordinary events from the CMCC model. Each panel shows ordinary events (gray), the top 10 % data portion (in red), and the linear fit (black dashed line) of that top portion in randomly selected locations.*

**Table S1.** *Spatial categories and location of sample series selected to illustrate the Weibull fit to ordinary events.*

| Series | Climate | Roughness | Topography | Longitude | Latitude |
|--------|---------|-----------|------------|-----------|----------|
| Series 1 | Co | R5 | T4 | 6.9836 | 45.2795 |
| Series 2 | Ar | R3 | T1 | 0.5996 | 41.4725 |
| Series 3 | Ar | R3 | T1 | 16.0656 | 41.1755 |
| Series 4 | Tm | R3 | T2 | 15.6856 | 44.2805 |
| Series 5 | Co | R5 | T3 | 12.0756 | 46.7375 |
| Series 6 | Co | R5 | T3 | 6.1096 | 45.7115 |

| Series 7 | Co | R5 | T3 | 5.4636 | 44.7665 |
|----------|-----|-----|-----|---------|---------|
| Series 8 | Tm | R3 | T2 | 0.5236 | 43.0655 |
| Series 9 | Co | R5 | T2 | 14.5076 | 46.5215 |
| Series 10 | Tm | R3 | T2 | 11.6956 | 42.9845 |
| Series 11 | Td | R5 | T4 | 13.4056 | 47.0615 |
| Series 12 | Co | R5 | T2 | 15.0016 | 46.6025 |

Finally, on the bootstrap this analysis quantifies spatial sampling uncertainty within categories (Sections 3.3.2-3.3.3). The narrow 95% confidence intervals in Figure 6.d-f indicate that category-level means are statistically precise, being robust to the specific random selection of the 100 points per category. This sampling uncertainty is smaller than the observed intra-category variability, which reflects fine geophysical heterogeneity (Section 5.4 and **comment R1GC2**).

Based also on **comment R1SC17**, we have replaced lines 315-319 to further develop this method with : *"... A bootstrap resampling was used to derive 95% confidence intervals for ensemble mean return levels (U_50) for each disaggregated spatial category level shown in Figure 6 (Tibshirani and Efron, 1993). Ensemble means were first calculated at each point by averaging the three CPM return level estimates. Then, for each spatial category level (e.g., Climate "Temperate" - Tm), all points from spatial categories containing that characteristic were aggregated, and the confidence intervals were computed by spatially resampling the ensemble means from all points within the category (with replacement, 1000 iterations), computing the mean of each bootstrap sample, and extracting the 2.5th and 97.5th percentiles. This quantifies spatial sampling uncertainty within categories."*

**Specific comments**

**R1SC1 Line 8 - 9:** After reading a bit further, I realized that this sentence is quite confusing for several reasons. You summarize your complete methodological package into one short sentence, which I think needs elaboration. I would rework this to make things more clear. Three reasons why it is unclear:

- The term 'surface based spatial categorisation' is vague when no additional information is added. Surface on itself can refer to many things (e.g. also surface winds) and only later

you explain that you classify into terrain, roughness and climate zones, I think you can be a bit more elaborate and specific already in the abstract.

- The way the sentence is written now, the reader might think you used PCA for the spatial categorisation.
- You mention PCA, a very widely used and general method, but do not explain here what you use it for in your work, so also here I advise to be a bit more specific and elaborate.

**R//:** Thank you for your comment. We have now changed lines 8-10 to: *"... in central Europe. We categorise locations by climate, roughness, and topography; then use Principal Component Analysis to quantify inter-model agreement and divergences; and apply the Simplified Metastatistical Extreme Value (SMEV) method to the 10-year simulations to estimate 50-year return levels (U_50) required for wind turbine design. Our results…"*

**R1SC2 Line 26-27:** This sentence is unclear to me. It could be because regional variability is mentioned twice, but in different ways, so I think the sentence can be more concise.

**R//:** Thank you for noticing this redundancy. To improve this section, we have replaced lines 26-28 with: *"...uncertainties (Prein et al., 2015; Fosser et al., 2024). Regional variability in extreme wind changes under future scenarios exhibits both intensification and attenuation depending on season and location (Pryor et al., 2020; Outten and Sobolowski, 2021; Ma et al., 2025), underscoring the need for high-resolution climate modelling in wind energy planning and assessment…"*

**R1SC3 Line 31:** A few comments

- Please distinguish between GCM and RCM resolutions, as GCMs have a nominal horizontal grid increment of 100 km, not 10 km.
- Using a term like "horizontal grid spacing" might also be a better idea than resolution, because the effective resolution is many times lower than the grid spacing. You could also clarify with a sentence that you are referring to grid spacing, not effective resolution.
- Please be more specific with what you mean with "general" wind patterns. GCMs only capture synoptic and larger scales. RCMs (dx > 10 km) also capture some mesoscales. Perhaps it's better to describe which scales both of them do not capture, that the CORDEX FPS runs do capture.
- Also, I recommend changing "cannot capture convective motions" to "cannot explicitly resolve…"

**R//:** We thank the referee for this clarification, which has contributed to the rigour of the ideas we express. Following also the response to **comment R2SC1**, we have changed lines 31-32 to: *"...global and regional climate models (GCMs and RCMs) have served as essential tools for understanding large-scale atmospheric flows. However, while GCMs (∼100 km grid spacing) capture synoptic scales and RCMs (∼10 km) extend into larger mesoscales, neither can explicitly resolve convective processes and represent fine-scale surface heterogeneity that affects local wind behaviour and extremes (Soares et al., 2012; Prein…"*

**R1SC4 Line 55-64:** This entire section is about methods you will not use eventually. Therefore, I would suggest to shorten it a bit and also to merge it with the section where you introduce SMEV, the method that you will be using. More generally, I would check where else in the introduction you could make the sequence more logical , because I feel like you switch back and forth between surface characteristics, inter model comparison and EV estimation quite a bit, which is a bit difficult for the reader.

**R//:** We appreciate the reviewer's suggestion to improve the introduction's flow and structure. GEV/GPD methods are the current standard in wind energy extreme-value analysis, and it is important to address their limitations (lines 55-64) to justify adopting SMEV within the wind energy community. To shorten and improve the logical sequence as suggested by the reviewer, we have restructured lines 55-80. First, we face the lack of long-term observations by conducting a relative intercomparison of the models using PCA. Then, we combined the discussion of extreme value methods (GEV/GPD limitations and SMEV advantages) into a single paragraph. This change removes repetitive references to "short time series". Next, we introduced spatial categorization as a complementary approach to address surface differences. This reorganization keeps the necessary context while making the text easier to read. The revised structure is as follows: CPM capabilities, the challenge of extreme-value estimation from short records, SMEV as a solution, and spatial categorization of surface effects.

Lines 55-80 have been replaced by: *"An effective use of CPMs for wind energy requires addressing key methodological challenges. First, the limited availability of long-term wind observations at hub heights restricts direct and absolute validation of CPM performance. We cover this by comparing models using Principal Component Analysis (PCA) to break down the variance structure into consensus signals and systematic differences. This method provides a relative assessment that shows where CPMs agree, which builds confidence in climatological patterns, and where they differ, which highlights structural uncertainties.*

*Second, estimating extreme return levels like U_50 from short CPM simulations, usually spanning 10-20 years, brings statistical challenges. Traditional extreme value methods, like Generalised Extreme Value (GEV used for block maxima) and Generalised Pareto Distribution (GPD, used for threshold exceedances), are well-documented in wind studies (Donat et al., 2010; Pfahl, 2014; Harris and Cook, 2014; Ma et al., 2025). However, these methods are sensitive to record length, making parameter estimation less reliable with shorter time series (Larsén et al., 2015; Coles et al., 2001). We address this issue using the Simplified Metastatistical Extreme Value (SMEV) approach (Marra et al., 2019). SMEV looks at the entire distribution of independent events rather than just annual maxima or threshold exceedances, making it more reliable for limited time series (Dallan et al., 2024). SMEV has been successfully used for extreme precipitation from in-situ observations, remote data, and CPM simulations (Miniussi and Marra, 2021; Dallan et al., 2023; Correa-Sánchez et al., 2025a; Marra et al., 2022). Here, we apply SMEV for the first time to wind speed analysis, expanding its use to wind energy applications.*

*Third, surface variability influences wind patterns at hub heights due to interactions between the surface and the atmosphere. Climate types, surface roughness, and topography affect boundary*

*layer dynamics through differential heating, roughness effects, and turbulence generation (Courault et al., 2007; Avissar and Schmidt, 1998). These factors impact winds at a height of 100 meters. Therefore, we create a spatial categorisation framework based on climate, roughness, and topography. This framework allows for a systematic assessment of CPM performance across different surface conditions relevant to wind energy."*

**R1SC5 Line 68-69:** This sentence is a bit confusing for me, as I don't see how the parts are connected. With addressing the limitations of observations with an inter model comparison, do you mean that models provide spatial fields that observations cannot and therefore capture the spatial variability of signals? That would be valid, but I don't understand why the part of performing a PCA-based inter-model comparison is merged into that sentence. Perhaps elaborate and/or split it up into more simple chunks.

**R//:** Thank you for pointing this out. Following your suggestion and in the same line as the response for **comment R1SC4,** we have elaborated and broken down this idea now starting in line 55 to provide accuracy and clarity: *"… An effective use of CPMs for wind energy requires addressing key methodological challenges. First, the limited availability of long-term wind observations at hub heights restricts direct and absolute validation of CPM performance. We cover this by comparing models using Principal Component Analysis (PCA) to break down the variance structure into consensus signals and systematic differences. This method provides a relative assessment that shows where CPMs agree, which builds confidence in climatological patterns, and where they differ, which highlights structural uncertainties.*

*Second, estimating extr…"*

**R1SC6 Line 91-96:** After reading the manuscript I went back to these objectives. I think that it could help to be a bit more explicit that objective 1 is a full domain analysis based on spatial maps. And that objective 3 and 4 build on objective 2 to perform analyses for the different categories, because the link is not very clear the way it is written now.

**R//:** We thank your comment, and we modified the text in order to provide clarity to the structure of the ideas in lines 91-95 as follows: *"...approach with four key objectives. First, we conduct a domain-wide analysis examining inter-model differences in extreme wind representation across the full study area. Second, we establish spatial categories based on climate, surface roughness, and topographic features to account for heterogeneous surface-atmosphere interactions that modulate wind patterns. Using this categorisation framework, our third objective quantifies spatio-temporal agreement among the three CORDEX-FPS CPMs in simulating turbine-height wind speeds across different surface conditions. Fourth, we apply SMEV to estimate extreme wind speeds from short simulations for each spatial category. We then discuss …"*

**R1SC7 Line 113-116:** Model physics (e.g. convection scheme) and stochastic variability are indeed an important part of why the CPMs might differ, but I think that the following elements can also be important:

- Model numerics: e.g. the number of vertical levels used and/or the amount of numerical diffusion applied, .... Could you add something about it to the text?
- Simulation protocol: continuous vs. chunked simulations, temporal spin up. Perhaps the models had the same settings here ? Could you add some notes on this?
- Also, next to the convection scheme, also the parametrization of the planetary boundary layer (PBL) is important in shaping low level wind variability. The CPMs might be using different ones. Please mention this too. It would be helpful to add into the table the convection & PBL schemes used by the different models. Or at least to give a general idea of how they differ between models?

**R//:** We thank the reviewer for their comments, which certainly contribute to the rigour of our work. We have supplemented the Section 2 with information available in the original papers of each model, previous work on the nature of these models, and the CORDEX-FPS project documentation paper. From these sources, and following your suggestions we have provided more details on the configuration of this type of simulations (free-running) and PBL schemes in Section 2 (Table 1 and line 115):

*"...subsequently nested within these RCMs. The models rely on two atmospheric cores: CMCC and ETH use the COSMO model, while CNRM uses AROME. For representing the planetary boundary layer (PBL), CMCC and ETH use the TKE-based scheme (Mellor and Yamada, 1982; Raschendorfer, 2001), while CNRM employs the CBR turbulence scheme (Cuxart et al., 2000; Bougeault and Lacarrere, 1989) with PMMC09 for shallow convection (Pergaud et al., 2009). The models have different vertical levels (ETH: 60 levels, CNRM: 60 levels, CMCC: 50 levels) and horizontal diffusion methods: CMCC uses fourth-order Smagorinsky hyper-diffusion, ETH has no explicit horizontal diffusion, and CNRM uses semi-Lagrangian horizontal diffusion (Adinolfi et al., 2020; Leutwyler et al., 2016; Caillaud et al., 2021; Correa-Sánchez et al., 2025). Even though CMCC and ETH share the COSMO framework and PBL scheme, they differ in computational implementation (ETH: GPU-accelerated COSMO; CMCC: standard COSMO-CLM), horizontal resolution, vertical levels, and horizontal diffusion methods \citep{Leutwyler2016, Adinolfi2020}, which may affect how they represent convective processes. All models follow the CORDEX-FPS protocol with continuous free-running simulations. Table 1 summarises model cores, convection treatment, PBL schemes, vertical levels, and diffusion methods.To enable direct comparison…"*

*Table 1.* *CPM members used for inter-model agreement assessment, showing reference names, original resolutions, coupled RCM configurations, and key technical specifications. All models were remapped to a common 3 km grid for comparison.*

| Institute | CPM | RCM | Atmospheric Core | PBL Scheme | Vertical Levels | Horizontal Diffusion |
|-----------|-----|-----|------------------|------------|-----------------|----------------------|
|           |     |     |                  |            |                 |                      |

| CMCC Euro-Mediterranean Center on Climate Change | CCLM 3 km (Adinolfi et al., 2020; Rockel et al., 2008) | CCLM 12 km (Adinolfi et al., 2020; Rockel et al., 2008) | COSMO | TKE-based (Mellor and Yamada, 1982; Raschendorfer, 2001) | 50 | Fourth-order Smagorinsky hyper-diffusion |
|---|---|---|---|---|---|---|
| CNRM Centre National de Recherches Météorologiques | CNRM-AROME41t1 2.5 km (Caillaud et al., 2021) | CNRM-ALADIN63 12 km (Nabat et al., 2020) | AROME | CBR (Cuxart et al., 2000; Bougeault and Lacarrere, 1989) | 60 | Semi-Lagrangian horizontal diffusion (SLHD) |
| ETH Institute for Atmospheric and Climate Science | COSMO-crCLIM 2.2 km (Leutwyler et al., 2016; Rockel et al., 2008) | COSMO-crCLIM 12 km (Leutwyler et al., 2017; Rockel et al., 2008) | COSMO (GPU-accelerated) | TKE-based (Mellor and Yamada, 1982; Raschendorfer, 2001) | 60 | None (implicit) |

We also added a new paragraph in line 481 in Section 5.4, to deep in to the discussions of these differences in the estimation of extreme winds:

*"...The CPM ensemble analysed here differs in multiple aspects of model configuration: dynamical cores (COSMO vs AROME), PBL parameterisations (TKE vs CBR), vertical discretisation (50-60 levels), horizontal diffusion approaches (fourth-order Smagorinsky vs none vs SLHD), and horizontal grid spacing (2.2-3.0 km), as explained in Section 2. A key finding from our inter-model comparison is that CMCC and ETH show U_50 differences that are similar in size to those between models using different PBL schemes, even though they have the same PBL parameterisations. For instance, the U_50 spread between CMCC and ETH (both using TKE-based PBL) is similar to the spread between CMCC and CNRM (TKE vs CBR). This pattern suggests that the uncertainty in extreme wind estimates between models comes from the combined effects of various configuration choices, including but not limited to PBL parameterisation and the fine scale heterogeneity. These inter-model differences highlight the importance of using ensemble-based approaches for reliable extreme wind assessment in wind energy applications …"*

**R1SC8 Fig 2:** If the 25 th percentile of the relative frequency of the categories is used, How can 35 out of 51 categories have relative frequency below this value? That does not agree with the definition of percentiles.. Please clarify in the text so that the reader can better understand this.
**R//:** We thank the reviewer for identifying this inconsistency in our description.

We have changed the manuscript to accurately describe the filtering approach. The proposed method selects the 17 most spatially abundant categories, which collectively represent 97.7% of the domain's pixels, while the 35 less frequent categories collectively contribute less than 2.3% of the domain's pixel coverage. In this way, we apply a coverage-based selection that ensures the focus on predominant conditions while maintaining the major spatial representativity.

We have corrected the description throughout the manuscript, i.e., in the Figure 2 caption by: *"...g strategy. a) Distribution of pixel relative frequencies across all 52 spatial categories ranked by abundance. Blue bars indicate the 17 most spatially abundant categories retained for analysis, which collectively represent 97.7% of the domain. Red bars show the 35 less frequent categories (collectively 2.3% of the domain) excluded to focus on predominant conditions. The log scale allows…"* and legend of Figure 2.a with: *"Coverage-based threshold"*.

Finally, we also modified the Methods section lines 191-192, to accurately reflect this coverage-based selection approach we implemented by: *"...prevalent spatial combinations. The categories were ranked by their relative frequency within the domain, and the 17 most spatially abundant categories were selected, collectively representing 97.7% of the domain's pixels (Figure 2). The 35 less frequent categories, which collectively contribute less than 2.3% of domain coverage, were excluded to focus on predominant conditions. This coverage-based selection maintains an adequate representation of the diverse…"*

**R1SC9 Line 208:** I think "temporal variability" should be changed to "temporal co-variability", because you do not perform the PCA on the individual time series.

**R//:** We thank the reviewer for pointing out this terminological issue, we have corrected it as suggested with *"temporal co-variability"* since we evaluate how the winds at 100 m from three different models vary together over time.

**R1SC10 Line 210:** What do you mean with "fluctuate"? If you mean "why models differ", then I would not use the word fluctuate. If you mean some form of fluctuation, please be more specific about what you mean.
**R//:** Thank you to the reviewer for highlighting this important detail, we changed in "*differ*" since more appropriate

**R1SC11 Line 224:** under identical atmospheric conditions ". I don't think this is correct because the model itself creates the atmospheric conditions in the domain . What I think you can say is "under identical atmospheric boundary conditions".
**R//:** We thank the reviewer for this important technical clarification, also highlighted in the **R2SC8 comment**. It is correct that CPMs develop their own internal mesoscale meteorology rather than

operating under strictly *"identical atmospheric conditions"*. While all three models share consistent initial and lateral boundary conditions from ERA-Interim, each of them develops model-specific atmospheric states through their distinct physics designs, parameterisations, and dynamical cores, which is precisely the "added value" of convection-permitting resolution (Prein et al., 2015; Coppola et al., 2020). To be clear about what we mean, we modified lines 123-124 with: *"...At each grid point, the three time series represent the same meteorological variable (wind speed at 100 m) but simulated by different models under identical atmospheric boundary forcing (ERA-Interim), enabling PCA to decompose the pure inter-model covariance structure…"*. This clarification better reflects that our inter-model comparison isolates physics-driven differences under equivalent boundary forcing.

**R1SC12 Section 3.2.2:** I agree that this seasonal correlation analysis is a nice addition to the PCA. Yet, you perform this on the absolute monthly maximum- could it not be informative to also perform this analysis on, for example, the 99th percentile? I expect that the P99 wind speed value is also very relevant for wind energy planning and will be less "noisy" than the monthly maximum, sort of a "less anomalous, but still very extreme" wind speed. You could add his, or you could add a good argumentation about why the analysis of both is equivalent.

**R//:** We appreciate this suggestion and have added seasonal correlation analysis using the 99th percentile (p99) of all hourly wind speeds within each month. For each calendar month, p99 was calculated from all the hourly values that month. This provides a more robust metric compared to the single monthly maximum while remaining focused on extreme values.

Although the results are generally similar to those obtained with the monthly maxima, we have revised and supplemented all the texts that are in line with the new Figure 5, which is based on a more robust metric.

In this sense, and in relation to the answer to **comment R2SC11,** the modifications we have included in lines 361-367 are: *"...The seasonal correlation results (Figure 5) show that the CNRM-CMCC pair exhibits generally lower correlations than the ETH-CNRM and ETH-CMCC pairs across most months and spatial categories, consistent with the PC2 divergence pattern, in which CNRM and CMCC systematically diverge while ETH remains near consensus across spatial categories. The spatial categories show modest influence on correlation magnitudes (typically ~0.1-0.3 variation), with high-elevation complex terrain categories exhibiting slightly lower correlations during the winter and transitional periods. We also performed this same analysis with the monthly maximum, and the correlational patterns obtained were similar, yielding consistent model rankings and seasonal patterns.…"*

[Figure]

**Figure 5**. *Monthly 99th percentile wind speed correlation coefficient per spatial category and CPMs couple.*

**R1SC13 Line 268:** "Ordinary events" you say this is the entire set of independent events, but it is not clear what these independent events means at this point. In section 3.3.1 it becomes clear that these ordinary events are extracted local maxima separated with your correlation window. Therefore, I would be more careful with introducing the term already here because it can be confusing to the reader If you do introduce it here, please explain what it means.

**R//:** Thank you to the reviewer for pointing out this passage where the writing may be confusing to the reader. We agree that the term *"ordinary events"* was introduced without sufficient explanation of what is an *"independent event"* in this context. To amend this we have modified the line 268 with: *"The Simplified Metastatistical Extreme Value (SMEV) approach, proposed by Marra et al. (2019), relies on a different set of assumptions with respect to classical extreme value theory, which enables one to use the entire set of independent events (temporally uncorrelated local maxima termed "ordinary events", see Section 3.3.1) instead of annual maxima or peaks over threshold only."*

**R1SC14 Line 271:** "instead of assuming this is infinitely high"- why would an algorithm assume that the frequency of occurrence of events is infinitely high? Can you please explain this part better please?

**R//:** The reviewer is correct that the expression *"infinitely high"* is confusing and we thank them for bringing this to our attention. Our intention was to contrast SMEV with classical extreme value theory, which relies on the asymptotic assumption of having an infinite number of occurrences in each block (usually, the year). We have changed the line 271 by : *"...available independent observations and explicitly accounts for their finite annual occurrence frequency, avoiding the asymptotic assumptions of classical extreme value theory (Marra et al., 2019). Therefore, SMEV accounts…"*

**R1SC15 Line 272:** When you say that it accounts for "intensity" do you mean that it takes into account the intensity distribution? Because "intensity" seems to refer to one specific intensity value. Please clarify this a bit more in the text.

**R//:** We thank the reviewer for this clarification request. To correct this we have modified line 272 with : *"...Therefore, SMEV accounts for both the distribution of wind speed magnitudes (through shape and scale parameters) and the annual frequency of occurrence of the events, thereby providing a more direct physical interpretation of …"*

Furthermore, to achieve greater terminological consistency throughout the text, we have changed line 293, which reads *"low-intensity events"*, to *"low-magnitude events"*, and line 474: Change *"different typical intensities"* to *"different typical magnitudes"*.

**R1SC16 Section 3.3.2:** This comment links back to my last general comment. Normally, Weibull distributions are used to fit an entire wind speed distribution, but you already filter for local maxima and then fit a Weibull distribution Even for full wind speed timeseries, Weibull distributions don't always work well for complex sites and I am also surprised that here it is applied to a maxima filtered timeseries , with the "left censoring" added to it . I think it is necessary to convince the reader (visually) that this fitting works. Could you either add a few supplementary plots or refer to some other work where this is clearly demonstrated?

**R//:** We are grateful to the reviewer for pointing this out, which is certainly in line with the response to the previous comment. Here, we would like to clarify two aspects:
- First, yes, the Weibull distribution is fitted to the top 10% ordinary events (independent wind peaks identified via decorrelation), not the complete hourly time series. Our ordinary events are independent high-wind peaks that collectively define the empirical distribution of extreme winds, whose upper tail is characterised by Weibull parameters estimated through left-censoring.
- Second, left-censoring is not a threshold filter. Following Marra et al. (2023), all ordinary events remain in the analysis: the lower 90% inform cumulative probability as "non-exceedances", while the upper 10% are used to estimate Weibull parameters. So we are not discarding or filtering data below a threshold.

In this sense, and similar to the response in **comment R1GC3**, we modified the text to clarify on these aspects by adding in line 275: *"...is and Cook, 2014; Tuller and Brett, 1984; Tye et al., 2014;*

*Hennessey Jr, 1977; Palutikof et al., 1999) and has been shown to effectively approximate distributions with theoretical foundations in atmospheric physics (Harris and Cook, 2014)...."*

Also, we added in line 301: *"...ordinary events in different locations. To illustrate, Weibull probability plots from 12 randomly selected locations of each CPM are provided in the supplementary material (Figures S4-S6; Table S1 shows their location and the spatial category to which they correspond). The linear relationship in ln-ln space validates the Weibull distribution assumption for the upper tail across diverse geographical and climatic conditions."*
The figures and tables mentioned in this reply are also the same as those found in the reply to **comment R1GC3.** To avoid repetition here, they have been left only in that first comment.

**R1SC17 Line 315:** Can you be more explicit what you bootstrapped? Is it the ordinary event dataset for each model? Is it the different points in each category? This is currently not very clear from this small section. Please clarify in the text.
**R//:** Thanks to the reviewer for pointing out the need for this methodological clarification. We bootstrapped the ensemble mean return levels (not the ordinary event datasets) within each spatial category. Specifically, we first calculated the ensemble mean return level at each sampled point by averaging the three CPM estimates. Then, for each spatial category, we performed spatial bootstrap resampling by randomly selecting ~100 ensemble mean values with replacement from the ~100 points within that specific category. This process was repeated 1000 times, and for each iteration we calculated the mean of the bootstrap sample. The 95% confidence intervals were then derived as the 2.5th and 97.5th percentiles of the resulting 1000 means.

We have now incorporated in the revised text explaining how the bootstrap resampling was performed on the collection of ensemble mean return levels across all points within each spatial category by replacing lines 315-319 with: *"... A bootstrap resampling was used to derive 95% confidence intervals for ensemble mean return levels (U_50) within each spatial category (Tibshirani and Efron, 1993). Ensemble means were first calculated at each point by averaging the three CPM return level estimates. Then, for each spatial category, the confidence intervals were computed by spatially resampling the ensemble means from all points within the category (with replacement, 1000 iterations), computing the mean of each bootstrap sample, and extracting the 2.5th and 97.5th percentiles. This quantifies uncertainty due to spatial sampling variability within categories."*

**R1SC18 Line 316:** for each disaggregated layer " I think this is the first time you use the term "layer" and it is unclear what you are referring to. Could you please clarify in the text?
**R//:** We appreciate you pointing this out to us. In relation to **comment R1SC17**, we have changed the term *"layer"* in line 316 to *"spatial category"*.

**R1SC19 Figure 6:** Can you add a physical interpretation in the text on why the lowest roughness class, i.e. open water areas, is characterized by the lowest U50 wind speed estimates? Because low roughness corresponds to less frictional deceleration so in this sense you would expect higher extreme wind speeds. Are there perhaps less summer convective storms over water? (I don't know immediately myself).

**R//:** Thank you for this observation. The lower 50-year return-level estimates for water surfaces (R1: 23 m/s vs R3-R5: 25-27 m/s) arise from the combined effects of sample composition, physical processes, and SMEV parameterisation, rather than from a single dominant factor.

**1.** The geographic classification of the 100 R1 sampling points reveals that 93% are located in semi-enclosed seas (Mediterranean 63%, Adriatic 30%), with only 7% in inland lakes. This distribution is critical for interpreting results: R1 represents Mediterranean and Adriatic waters with inherently limited fetch (~600-800 km), not "true offshore" in open ocean conditions. Semi-enclosed basins are subject to fetch-dependent wind development constraints that limit extreme wind magnitudes compared to unlimited-fetch offshore environments where $U\_50$ values of 12-15 m/s are typically observed in wind energy assessments (Petersen, 1997).

**2.** The SMEV return level formulation $U\_T = \lambda * [n * \ln(T)]^{(1/k)}$ reveals differential parameter sensitivity. Our results (new Fig. 7) show 18-20% difference in scale parameter ($\lambda$: R1 = 8.8 m/s vs R3-R5 = 10.4-11.1 m/s) but only 3-5% in shape parameter (k: R1 = 2.36 vs R3-R5 = 2.42-2.47). Following the sensitivity framework of Marra et al. (2019) and formal uncertainty propagation analysis of Siena et al. (2023), $\lambda$'s approximately linear influence dominates over k's logarithmic effect (through exponent 1/k), such that the 18% lower $\lambda$ translates proportionally to lower $U\_50$ estimates while the small k difference produces only ~3% amplification. This explains why comparable annual maxima (Fig. 3a-c) yield different $U\_50$ estimates (Fig. 6): water bodies experiencing occasional intense synoptic events (25-28 m/s) matching terrestrial peaks but lacking sustained convective tail enrichment exhibit lower $\lambda$ despite similar maxima.

**3.** As we mentioned in the manuscript, the terrestrial categories (R3-R5) produce convectively-driven extreme winds that CPMs explicitly resolve, including deep convective downdrafts, gust fronts, and orographic acceleration (Prein et al., 2015; Manning et al., 2022). These processes produce frequent, high-magnitude events that populate the Weibull tail, thereby elevating $\lambda$ in the land. Over water, atmospheric stabilisation by cold sea surfaces suppresses deep convection, limiting extreme wind generation primarily to synoptic forcing (such as cyclonic systems). While these synoptic events can produce intense individual maxima, they lack the systematic high-frequency tail enrichment that convective processes provide on land.

**4.** There is a previously known CORDEX-FPS CPMs limitation in representing air-sea interactions. As documented by Caillaud et al. (2021) for the CNRM-AROME model: *"For the surface boundary conditions, we impose the Sea Surface Temperatures (SSTs) from interpolated monthly ERA-Interim SSTs (around 80 km) as recommended in the CORDEX-FPS on Convection simulation protocol... A step forward to take advantage of a higher resolution SST would be to have a high-resolution high-frequency coupling between atmosphere and ocean to represent the complex air-sea interactions, but it is not yet affordable in long climate simulations".* which creates a scale mismatch: CPM atmosphere at 2-3 km resolution interacting with coarse-resolution (appx 80 km) monthly-mean SSTs.

These limitations had also been highlighted by Prein et al. (2015) and are persisting in CORDEX-FPS protocols (Caillaud et al., 2021; Coppola et al., 2020). This may lead to lower estimates of extreme winds over water.

We have therefore added in the line 312 of the manuscript: *"...10-year simulation periods.*

*It should be noted that SMEV return levels are more sensitive to the scale parameter λ (approximately linear relationship: δq/q ≈ δλ/λ) than to the shape parameter k (logarithmic dependence through exponent 1/k; Marra et al., 2019; Siena et al., 2023). This differential sensitivity is critical for interpreting inter-category differences (see Fig. S7).*

*The estimation procedure was …"*

We have also included new lines in the text referring to this supplementary information in the Results section, specifically in line 392:
*"...*
*To complement the U_50 estimations in Fig. 6 we have analysed the distribution of the Weibull parameters across the disaggregated spatial categories (Figure S7). The lower U_50 estimates in R_1 primarily reflect an 18% lower scale parameter λ (median: 9.1 m/s vs 10.4-11.1 m/s in R_3-R_5), while shape parameter k shows minimal variation (3-5%; Fig. S7).*

[Figure]

***Figure S7.*** *Distribution of Weibull parameters across disaggregated spatial categories. Panels (a-c) show shape parameter k. Panels (d-f) show scale parameter λ [m/s]. Each boxplot represents 100 locations per category. The dashed line in (a-c) marks k=2 (Rayleigh distribution); median values are annotated within boxes.*

*…"*

Also, in the Discussions we have at the end of section 5.5 in line 505:
*"...*
*Our U_50 estimates over R_1 should be interpreted cautiously for offshore wind energy applications. First, considering our study area, the R_1 sample represents semi-enclosed Mediterranean/Adriatic waters rather than the unlimited-fetch open-ocean conditions characteristic of major offshore wind farms in other open-basin seas, such as the North Sea, where persistent westerly flow and unrestricted fetch typically generate systematically higher extreme wind speeds. Second, CORDEX-FPS CPMs employ coarse-resolution SST forcing (~80 km monthly means), creating scale mismatches with the fine-resolution (2-3 km) atmospheric grid (Caillaud et al., 2021), which may contribute to conservative marine wind estimates. These geographic and technical factors, combined with the physical process whereby water surfaces experience occasional intense synoptic events but lack the high-frequency convective events enriching terrestrial tail distributions, result in 18\% lower scale parameter λ in R_1 compared to land categories R_3–R_5 (Fig.S7, translating approximately linearly to lower U_50 estimates.  …"*

**R1SC20 Conclusion:** I think it would be good to very briefly repeat some of the suggestions/limitations you mentioned here again. For example, the need for CPM evaluation for observations, measurement campaigns and the need to involve more CPMs. Also, I would recommend to use the past tense for summarizing the stuff that you did. For example "We first investigate → "First, we investigated…". General truths or findings can remain in present tense of course ( e.g. "This highlights the need for ensemble approaches..."). I personally think this makes more sense, but you can also leave it like this if you don't agree.

**R//:** We appreciate these valuable suggestions. We have incorporated your suggestion regarding timing into the text between lines 519 and 525, keeping it present for general truths or findings:

*"...at turbine heights. We first investigated whether there are discrepancies among the models and how well the models agree in representing winds at 100 m. We demonstrated substantial inter-model differences in extreme wind representation, with systematic variations exceeding 15–20 m s^-1 in annual maxima values in complex terrain despite identical geographic domains and atmospheric forcing. Furthermore, we provided a framework based on model inter-comparison for evaluating the CPM simulations of extreme wind for different spatial categories in the absence of long-term observational data. We then combined these simulations with the Simplified Metastatistical Extreme Value approach (SMEV) to assess extreme wind return levels at turbine heights.*

*We organise the analyses..."*

To briefly summarise the limitations, we have added the following to line 546: *"...are fundamental.*

*Key limitations include the lack of observational datasets that provide enough spatial and time coverage and that are easily accessible for direct validation. Since multi-decadal simulations are useful for planning, turbine design, and investment strategies, there are other limitations regarding*

*the relatively short 10-year simulation period that includes only three ensemble members. Future efforts should focus on expanding observational networks at hub heights, using longer simulations with more ensemble members, and carrying out targeted measurement campaigns across relevant spatial categories.*

*CPMs projections…"*